# Meteorological Conditions Leading to the 2015 Salgar Flash Flood: Lessons for Vulnerable Regions in Tropical Complex Terrain

Carlos D. Hoyos [1,2], Lina I. Ceballos [1,2], Jhayron S. Pérez-Carrasquilla [1,2], Julián Sepulveda [1,2], Silvana M. López-Zapata [1,2], Manuel D. Zuluaga [1], Nicolás Velásquez [1,2], Laura Herrera-Mejía [1,2], Olver Hernández [2], Gisel Guzmán-Echavarría [1,2], and Mauricio Zapata [1,2]

[1]Universidad Nacional de Colombia, Sede Medellín, Facultad de Minas, Departamento de Geociencias y Medio Ambiente
[2]Sistema de Alerta Temprana de Medellín y el Valle de Aburrá (SIATA), Área Metropolitana del Valle de Aburrá (AMVA)

**Correspondence:** Carlos D. Hoyos (cdhoyos0@unal.edu.co)

**Abstract.** Flash floods are a recurrent hazard for many developing Latin American regions due to their complex mountainous terrain and the rainfall characteristics in the Tropics. These regions often lack the timely and high-quality information needed to assess, in real-time, the threats to the vulnerable communities due to extreme hydrometeorological events. The systematic assessment of past extreme events allows improving our prediction capabilities of flash floods. In May 2015, a flash flood in La Liboriana basin, municipality of Salgar, Colombia, caused more than 100 casualties and significant infrastructure damage. Despite the data scarcity, the climatological aspects, meteorological conditions, and first-order hydrometeorological mechanisms associated with La Liboriana flash flood, including orographic intensification and the spatial distribution of the rainfall intensity relative to the basin's geomorphological features, are studied using precipitation information obtained using a weather radar quantitative precipitation estimation (QPE) technique, as well as from satellite products, in situ rain gauges from neighboring basins, quantitative precipitation forecasts (QPFs) from an operational Weather Research and Forecasting (WRF) model application, and data from reanalysis products. La Liboriana flash flood took place during a period with negative precipitation anomalies over most of the country as a result of an El Niño event. However, during May 2015, moist easterly flow towards the upper part of La Liboriana caused significant orographic rainfall enhancement. The overall evidence shows an important role of successive precipitation events in a relatively short period, and of orography, in the spatial distribution of rainfall and its intensification as convective cores approached the steepest topography. There were three consecutive events generating significant rainfall within La Liboriana basin, and no single precipitation event was exceptionally large to generate the flash flood, but rather the combined role of precedent rainfall, and the extreme hourly precipitation triggered the event. The results point to key lessons for improving local risk reduction strategies in vulnerable regions with complex terrain.

## 1 Introduction

On the morning of May 18, 2015, at around 02:40 local time -LT- (UTC-5), a deadly flash flood in *La Liboriana* river basin inundated the town of *Salgar*, in the Department of *Antioquia*, Colombia (see Figure 1), killing more than one hundred people,

and leaving around 535 houses destroyed and significant public infrastructure damage[1]. From May 14 to May 18, several stratiform rainfall events and intense convective storms took place over the basin, triggering the flash flood. The reconstruction cost was estimated at approximately twelve million dollars[2]: about two and a half times the annual budget of the municipality[3].

Salgar, founded in 1903, and according to the national government[4] with a population of around 17600 people in 2015, of which 8800 reside in the urban area, is a typical complex terrain South American town settled in the Andes Cordillera, erected on the river margins of the main channel of La Liboriana watershed, near the confluence with El Barroso river, in a flash-flood susceptible region from a geomorphological perspective.

A large number of cities and municipalities in the Andean region, not only in Colombia but also in Bolivia, Ecuador, and Perú, are settled in small to medium size river basins similar to La Liboriana, in highly complex mountainous regions with hills exhibiting steep slopes, significant elevation differences relative to the river channel, and urban areas settled in areas prone to flash-flooding. In Colombia, for example, where the Andes Cordillera trifurcate into branches, 67% of the country's population lives in the Andean sub-region, which corresponds to 30% of the territory[5]. This condition increases the vulnerability of the country due to the fact that important human settlements frequently occupy floodplains.

Flash floods are associated with short-lived, very intense convective precipitation events, usually enhanced by the orography, over highly saturated land surfaces with steep terrains (Šálek et al., 2006; Llasat et al., 2016; Douinot et al., 2016; Velásquez et al., 2018). According to the US National Oceanic and Atmospheric Administration (NOAA), flash floods are triggered by heavy or excessive rainfall in a short period of time, appearing, in general, within six hours from the onset of torrential rainfall (Jha et al., 2012). However, it is important to state that flash floods are not only the result of the rainfall event immediately before the flooding (the triggering event); the spatio-temporal structure of the rainfall in the days prior to the extreme flooding (preconditioning events) also play an important role modulating the overall moisture in the basin and in the occurrence of flooding. Flash floods are highly destructive, often resulting in significant human and economic losses, making them one of the most catastrophic natural hazards (Jonkman, 2005; Roux et al., 2011; Gruntfest and Handmer, 2001). Jonkman (2005), based on information from the International Disaster Database, shows that between 1975 and 2001 a total of 1816 worldwide freshwater flood events killed over 175 thousand people and affected more than 2.2 billion people. These events not only caused human and economic losses but also damages to ecosystems and loss of historical and cultural values. In Colombia, there have been several flash flood events in the last decade associated with large-scale climate forcing patterns and with isolated extreme precipitation events. The 2010-2011 La Niña event triggered more than 1200 flooding events, affecting the lives of more than

---

[1]As reported by local and national media and the government: https://www.elcolombiano.com/multimedia/videos/salgar-tres-anos-despues-de-la-tragedia-KD8702862, https://caracol.com.co/emisora/2016/05/17/medellin/1463513573_945644.html, http://portal.gestiondelriesgo.gov.co/Paginas/Noticias/2015/Antecion-Emergencia-Salgar-Antioquia.aspx

[2]See the report in https://www.eltiempo.com/colombia/medellin/termina-reconstruccion-de-salgar-antioquia-tras-avalancha-107534

[3]See the annual budget for all municipalities in the Department of Antioquia: http://www.antioquiadatos.gov.co/index.php/8-2-2-1-1-presupuesto-definitivo-de-ingresos-de-los-municipios-de-antioquia-por-subregion-2015

[4]See projections at (accessed April 27, 2019):http://www.dane.gov.co/files/investigaciones/poblacion/proyepobla06_20/ProyeccionMunicipios2005_2020.xls

[5]Estimated based on the area of the Colombian sub-regions and government projections of population available at (accessed April 27, 2019):http://www.dane.gov.co/files/investigaciones/poblacion/proyepobla06_20/ProyeccionMunicipios2005_2020.xls

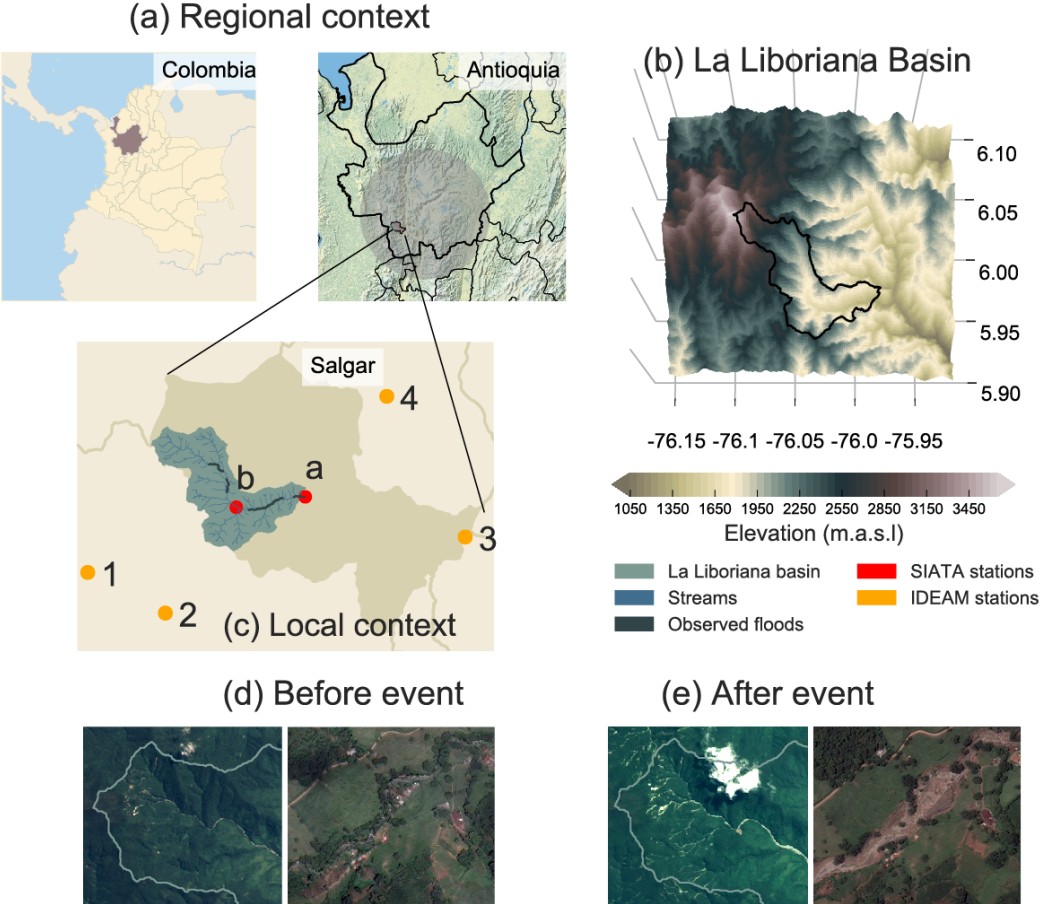

**Figure 1.** (a) to (c) Geographical context of the municipality of Salgar and La Liboriana basin. The panels present (a) the location of Colombia, the Department of Antioquia, and (b) and (c) Salgar, settled in the southwestern region of the Department, on the eastward facing hill of the western branch of the Andes Cordillera. The circular gray shadow denotes the 120 km radius area from the weather radar (C-Band) site. (b) Three-dimensional representation of La Liboriana basin using ALOS-PALSAR Digital Elevation Model (12.5 m resolution) highlighting the steepness of the region. Cerro Plateado corresponds to the highest hills of the basin at 3600 m.a.s.l. (c) Zoom to the municipality of Salgar and La Liboriana basin, showing the location of four rain gauges (1 to 4) from the Colombian National Weather Service (IDEAM), and two rain gauges (a and b) installed post-event by the Sistema de Alerta Temprana de Medellín y el Valle de Aburrá (SIATA). The IDEAM gauges correspond to codes 11020010 (1), 11025010 (2), 26210110 (3), and 26215010 (4). (d) y (e) Images before and after the Salgar May 18, 2015 flash flood event, showing an example of landslide scars in the upper basin, and erosion and changes in the delineation of the main channel. The first column corresponds to the pansharpened natural color composition from Landsat 8 images (path: 009, row: 056) with 15 m resolution for the upper basin ( Before: 2013-07-16, and After: 2017-05-24). The second column corresponds to Google Earth Pro V 7.3.2.5491, 2018 for the main channel. The images are from Digital Globe, 2018 (Before: 2014-10, After: 2015-05).

3 million people (around 7% of the country's population) and causing damages estimated in more than 6.5 billion US dollars (UN-CEPAL, 2012).

One of the most critical challenges associated with flash floods is their simulation and prediction with useful lead times (Yamanaka and Ma, 2017; Borga et al., 2011; Marra et al., 2017; Hardy et al., 2016; Ruiz-Villanueva et al., 2013). Skillful forecasts of the likelihood of the occurrence of a flash flood require a deeper insight into their triggering processes (Klemeš, 1983; Klemes, 1993; Merz and Blöschl, 2003). These processes are complex and controlled by a range of variables including the rainfall regime (Merz and Blöschl, 2003). Several authors have assessed the role of the spatio-temporal structure of rainfall on flash flood occurrence in different watersheds around the globe. For example, Doswell et al. (1996), Kahana et al. (2002) , Schumacher and Johnson (2005), Delrieu et al. (2005), Milelli et al. (2006), Borga et al. (2007), Norbiato et al. (2008), Fragoso et al. (2012), Berne and Krajewski (2013), Peters and Roebber (2014), Peters and Schumacher (2015), Gochis et al. (2015), Piper et al. (2016), Llasat et al. (2016), and Baltaci (2017) evaluated the observational evidence associated with flash floods from a climatological point of view and from case studies, pointing to the following important conditions prior to the occurrence of flash floods: (i) existence of distinct synoptic patterns days before the flood events leading to extreme precipitation, (ii) recurrence of convective systems, either organized as training convective elements or quasi-stationary convection as part of Mesoscale Convective Systems (MCSs), (iii) antecedent long-duration stratiform rainfall saturating the basin prior to intense convective events, (iv) orographic rainfall enhancement, and (v) modulation of local rainfall by large-scale climate patterns. Implicitly, these studies point to the need to examine in detail, and with a high spatio-temporal resolution, the rainfall structure in a basin of interest to better understand the flash flood triggering processes. In the tropical Andean region there are several factors that increase the likelihood of flash flood occurrence, including the high availability of moisture in the atmosphere, the north-south migration of the Intertropical Convergence Zone (ITCZ) and its role in setting the conditions for short-lived convective precipitation events with high intensities in small spatial scales (Mapes et al., 2003; Houze, 2004; Houze et al., 2015), and the orographic rainfall intensification strengthening localized deep convective cores (Poveda et al., 2007; Falvey and Garreaud, 2007; Mora et al., 2008; Bookhagen and Strecker, 2008).

In addition to being the primary flash flood triggering factor under current climate conditions, potential changes in extreme precipitation frequency under climate change scenarios could plausibly increase flash flood recurrence around the globe (Hapuarachchi et al., 2011; Field et al., 2012), and in particular in the Tropics, although the confidence in the projection of the magnitude of the precipitation changes is low (IPCC, 2014). Local in situ evidence suggests that, while there are no long-term trends in the yearly cumulative precipitation in the Department of Antioquia, short-lived events show long-term increments in intensity and frequency (Urán, 2016; Urán et al., 2019), with a substantial reduction of the return period of extreme events with implications for engineering and risk management.

As it is the case of La Liboriana extreme event, in many regions in the Tropics and around the developing world, watersheds prone to flash floods are usually located in rural mountainous areas, with scarce or non-existent real-time hydrometeorological information, imposing a challenge for their prediction, modeling and, consequently, optimal risk management (Marra et al., 2017). The use of quantitative precipitation estimation (QPE) tools based on ground-based weather radar and satellite-based information for flash flood applications could potentially offset the lack of in situ precipitation products in small poorly gauged

basins, becoming an important tool for the improvement of the state-of-the-art understanding of flash flood-related processes such as orographical enhancement of extreme rainfall and runoff generation (Creutin and Borga, 2003; Wagener et al., 2007). Additionally, radar information is also important since antecedent rainfall serves as a surrogate of soil moisture, and different authors have shown that antecedent soil moisture significantly modulates the occurrence of flash floods (Tramblay et al., 2012; Rodriguez-Blanco et al., 2012; Coustau et al., 2012; Wagener et al., 2007; Castillo et al., 2003). Notwithstanding the limitations of radar retrievals (Šálek et al., 2006; Hardy et al., 2016), modern approaches combining radar information, in situ precipitation data, and model simulations are promising (Braud et al., 2016). Velásquez et al. (2018) present a hydrological modeling framework for the reconstruction of La Liboriana flash flood assessing the runoff generation processes, concluding that the flash flood and the associated regional land-slides in the region were strongly influenced by the observed antecedent rainfall, recharging the gravitational and capillary storages in the entire basin.

The aim of this study is to document the climatological aspects, meteorological conditions, and first-order hydrometeorological mechanisms triggering the 2015 La Liboriana flash flood, including orographic intensification and the spatial distribution of the rainfall intensity relative to the basin geomorphological features. We focus on the aspects related to the recurrence of the event and highlight the key lessons that should be incorporated into local risk reduction strategies for other vulnerable regions with similar climate features and terrain complexity. In spite of the data scarcity, the systematic analysis of the observational evidence of the successive rainfall events triggering La Liboriana flash flood, and the evaluation of the output of limited-area numerical prediction models, constitute an interesting case of study to improve our understanding of the main hydrometeorological factors modulating the occurrence of these events and their likelihood of occurrence. This type of study is useful in the context of policy-making, not only for short-term early warnings but also as a planning resource for long-term risk management and resilience building strategies.

The present work is structured as follows. Section 2 describes the region of study, La Liboriana basin, as well as the information sources used in this analysis. The assessment of the overall climatological and meteorological conditions, and hydrometeorological mechanisms triggering La Liboriana flash flood is presented in section 3 using precipitation information derived from a weather radar QPE technique, as well as from satellite products, in situ rain gauges from neighboring basins, quantitative precipitation forecasts (QPFs) from an operational Weather Research and Forecasting (WRF) model application, and data from reanalysis products. Finally, the discussion and most important conclusions are presented in section 4.

## 2 Study region, data, and methods

### 2.1 Geographical context

The municipality of Salgar is located in the southwest of the Department of Antioquia, in the westernmost branch (*Cordillera Occidental*) of the Colombian Andes. Figure 1a and b show the geographical context of the municipality of Salgar and La Liboriana basin. The Figure shows Colombia, the Department of Antioquia, and the location of Salgar. Figure 1c, shows the three-dimensional representation of La Liboriana basin using ALOS-PALSAR (ASF-JAXA, 2011) Digital Elevation Model (12.5 m resolution), highlighting the steepness of the region. La Liboriana is an eastward facing basin with an area of 56 km$^2$;

its main channel originates in *Cerro Plateado* at around 3600 m.a.s.l., and its outlet is located at 1300 m.a.s.l., where it joins El Barroso river, and then drain to *Cauca* river. Figures 1d and e show an example of landslide scars in the upper basin, changes in the delineation of the main channel, and considerable erosion as a consequence of the extreme precipitation event that triggered the Salgar flash flood.

## 2.2 In situ rain gauges

Before the occurrence of the flooding event, there were no rain gauges available in La Liboriana basin, nor in the municipality of Salgar. Figure 1b shows the location of four rain gauges from the Colombian National Weather Service (IDEAM) in nearby regions to La Liboriana basin. The records from in situ gauges are available with daily resolution and are useful to characterize the rainfall during May 2015 in a climate context. Figure 1b also shows the location of two rain gauges installed after the flooding event by the Sistema de Alerta Temprana de Medellín y el Valle de Aburrá (SIATA), the local early warning system of the Department of Antioquia's capital, with the purpose of validating the rainfall estimates using QPE techniques.

## 2.3 C-Band radar QPE

In the absence of in situ rainfall records during La Liboriana flash flood, we use precipitation estimates based on an empirical QPE technique described in Sepúlveda (2016) and Sepúlveda and Hoyos (2017) using reflectivity fields from a 350kW C-band polarimetric and Doppler weather radar manufactured by Enterprise Electronics Corporation. The method uses radar reflectivity retrievals, and in situ disdrometer and rain gauge information to finally obtain precipitation. The method is multi-stage, and it is based on (i) a regression of the radar reflectivity into the in situ disdrometer reflectivity, and (ii) a regression between the in situ reflectivity and the gauge-measured rainfall intensity. Figure 1a (zoom 1) shows a 120 km-radius area from the weather radar (C-Band) installation site. The radar scanning strategy, which includes four plan position indicator sweeps (PPIs) at 0.5°, 1.0°, 2.0°, and 4.0°, and four range height indicator sweeps (RHIs), allows estimating precipitation information every five minutes with a spatial resolution of about 128 m using the 1.0° PPI. The QPE technique was validated for La Liboriana basin using hourly and daily information from the two in situ gauges installed after the flooding event. Figure 2 shows a correspondence between the hourly and daily cumulative precipitation estimated using the QPE technique and the precipitation registered in situ. The correlation between the hourly and daily rain gauge records and the QPE estimations are, respectively, 0.65 and 0.74. Similarly, the root mean square error and the mean absolute error for the hourly and the daily records are 3.8 and 2.2 mm, and 10.4 and 7.2 mm, respectively. The high correlations and relatively small hourly errors indicate, despite the evident overestimation relative to the in situ tipping-bucket gauge records (slopes in Figure 2 are 0.6 and 0.62), a high reliability of the derived precipitation based on radar reflectivity fields.

### 2.3.1 Convective and stratiform precipitation

Radar reflectivity fields are also used to describe the spatio-temporal evolution of the precipitation events leading to the La Liboriana flash flood and to assess the partition into its convective and stratiform portions, using the classification methodology

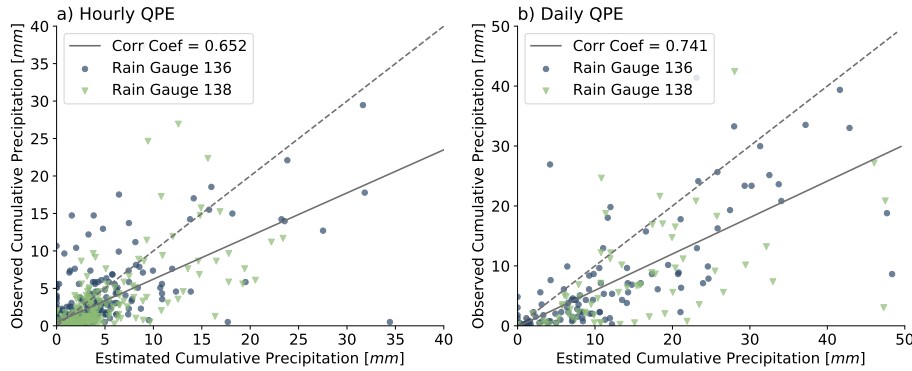

**Figure 2.** a) Scatter plot showing the relationship between the hourly cumulative precipitation estimated using the QPE technique and the in situ precipitation from the post-event SIATA rain gauges. The dashed line corresponds to a hypothetical perfect adjustment, and the continuous line to the actual linear fit between the QPE technique and the in situ precipitation. b) Similar to a) but for daily cumulative PPT.

proposed by Yuter and Houze (1997), and Steiner et al. (1995), which is based on the intensity and sharpness of the reflectivity peaks. Lightning activity is also analyzed together with radar reflectivity fields using data from in situ instruments that are part of a total lightning detection system developed at the University of Munich (LINET - Lightning NETwork), which locates cloud discharges and cloud-to-ground strokes (Betz et al., 2008, 2009). The efficiency in the detection of the two types of strokes is greater than 90%, identifying the time of occurrence with an accuracy of 100 ns.

### 2.3.2 Predominant propagation of precipitation events

In a complex-terrain setting, the orientation of the basin relative to the wind direction and to the predominant advection of the precipitation systems determines, to a significant extent, the potential for orographic enhancement. In order to evaluate this potential, the Li et al. (1995) methodology, based on the work by Tuttle and Foote (1990), is used to assess the prevailing direction in which precipitation systems approach La Liboriana basin. We estimated the velocity fields from precipitation retrievals (QPE technique) using a pattern cross-correlation technique and a variational approach, to satisfy continuity, for all the precipitation events over La Liboriana from May 2014 to August 2018. The dates of the precipitation events were obtained from the radar record. Precipitation events with an average rainfall intensity less than than 0.2 mm h$^{-1}$ were not considered.

### 2.4 Satellite-based precipitation products

In the case of La Liboriana flood, there is radar information available; however, for several basins around the world, the only spatially distributed precipitation information available in near real-time is from satellite retrievals. From a risk management point of view, it is beneficial to assess how useful the satellite information is to: (i) represent the spatio-temporal structure of the precipitation events triggering flash floods, and (ii) to issue advanced warnings to the public. For this assessment, we use the 3-hourly Tropical Rainfall Measuring Mission (TRMM) 3B42 v7 product, which is in good agreement with in situ

stations globally and regionally (TRMM, 2011; Kummerow et al., 1998; Huffman et al., 2007; Ceccherini et al., 2015), and the half-hourly Global Precipitation Mission IMERG v05 Final Precipitation (Huffman, 2017; Huffman et al., 2007; Joyce and Xie, 2011). The TRMM-3B42 v7 product has a 0.25° by 0.25° spatial resolution, and a spatial coverage spanning the entire

zonal band from 50°S to 50°N. The IMERG v05 Final Precipitation has global coverage and a spatial resolution of 0.1° by 0.1°. We also use GOES-13 4km spatial resolution infrared brightness temperatures (UCAR/NCAR-EOL, 2015) as a proxy for cold cloud tops, also useful to identify locations where precipitation is likely occurring.

  Although it is out of the scope of this research, it is important to highlight that extreme convection early detection and tracking algorithms based on combined satellite and radar information might also play a decisive role in risk management,

anticipating the threat to regions of interest. The reader is referred to the description of skillful Lagrangian tracking algorithms available in the literature (e.g. Carvalho and Jones, 2001; Handwerker, 2002; Vila et al., 2008; Zinner et al., 2008; Bellerby et al., 2009; Kober and Tafferner, 2009; Merk and Zinner, 2013).

## 2.5 Sea surface temperature and reanalysis products

Monthly sea surface temperature (SST) information from the NOAA Optimum Interpolation SST V2 (Reynolds et al., 2002),

with 1° horizontal resolution, is used to describe the climate context in which La Liboriana flood occurred and the long-term link between local precipitation and global SSTs. Similarly, atmospheric information (meridional and zonal winds, geopotential height and specific humidity) at different pressure levels is obtained from the ERA5 global reanalysis project with a spatial resolution of 30km (ECMWF, 2017).

### 2.5.1 Back trajectory analysis

Precipitation is, among others, a function of the net moisture influx into a region of interest. In the Tropics, the divergence/convergence of moisture fluxes and the migration of the ITCZ control the net moisture influx in a control volume. In long- (sustainable and resiliency-based urban/rural planning) and short-term (early warning) risk management applications, it is remarkably important to identify the moisture sources in a region to fully characterize the local hydrological cycle and the present and future potential hazards linked to extreme events. A Lagrangian approximation, such as the estimation of back

trajectories, is an optimal tool to identify the moisture sources for a region (Gimeno et al., 2012), allowing to characterize its climatology and to identify spatio-temporal patterns associated with the essential moisture sources in different time-scales (Dirmeyer and Brubaker, 2007; Drumond et al., 2008; Viste and Sorteberg, 2013; Drumond et al., 2014; Huang and Cui, 2015; Ciric et al., 2016; Stojanovic et al., 2017; Hoyos et al., 2018). The back trajectories, together with the specific humidity, are estimated using three-dimensional wind and specific humidity fields from ERA5 global reanalysis. These trajectories are an

excellent proxy for the origin of the moisture generating the observed precipitation events.

**Table 1.** WRF model schemes and parameterizations

|  | Schemes |
|---|---|
| **Microphysics** | Eta (Ferrier) |
| **Shortwave/Longwave Radiation** | Rapid Radiative Transfer Model (RRTMG) |
| **PBL** | Mellor-Yamada-Janjic |
| **Land surface** | Unified Noah land-surface model |
| **Surface** | Monin-Obukhov (Janjic Eta) |
| **Cumulus** | Tiedke scheme |

## 2.6   WRF Model

The WRF Model, version 3.7.1 (Skamarock et al., 2008) is the foundation for the operational numerical forecasts issued by SIATA on a daily basis. The model configuration uses three nested domains with 18 (191 x 191), 6 (82 x 118) and 2 (136 x 136) km horizontal resolution, and 40 vertical levels up to 50-hPa. The first domain (18 km), from 10ºS to 20ºN and from 60ºW to 90ºW, covers the entire geography of Colombia, the Caribbean Sea, the Colombian sector of the Pacific Ocean, and the Amazonia, in order to include the main external forcing factors of atmospheric circulation and precipitation over the territory. The second domain (6 km) includes the Andean region of Colombia (1º to 10ºN, and 72º to 78ºW). The third and last domain (2 km) is centered around the Aburrá Valley and extends from 5º to 7.5ºN, and 74.5º to 76.8ºW. The municipality of Salgar is located within the third domain. The model uses the output from the 12UTC 0.5º Global Forecast System (GFS) as initial and boundary conditions. The integration time step is 90, 60, and 10 s in the 18, 6, and 2 km resolution domains, respectively. Table 1 summarizes the schemes and parameterizations used.

## 3   Results

### 3.1   Climatology Context

La Liboriana flash flood occurred during May 2015, a month with multiannual average cumulative precipitation between 350 and 400 mm in the nearby region according to IDEAM records. The daily and cumulative rainfall recorded by the existing gauges is shown in Figure 3. Most of the rainfall (more than 85%) took place after May 12th, with the heaviest period between May 12th and May 22nd. The highest daily precipitation recorded by the rain gauges corresponds to accumulations around 55-60 mm. These daily values do not appear to be unusually high, corresponding to daily precipitation percentiles between 75th and 90th. This suggests that the available in situ records do not capture the unique spatio-temporal rainfall features that triggered the flash flood. While in situ rain gauges are perhaps the best method to measure precipitation intensity and total volume, their generally sparse nature limit their use in risk management applications, pointing to the need to combine information of local nature, or ground truth, with tools that provide spatio-temporal precipitation estimates such as satellites and weather radars.

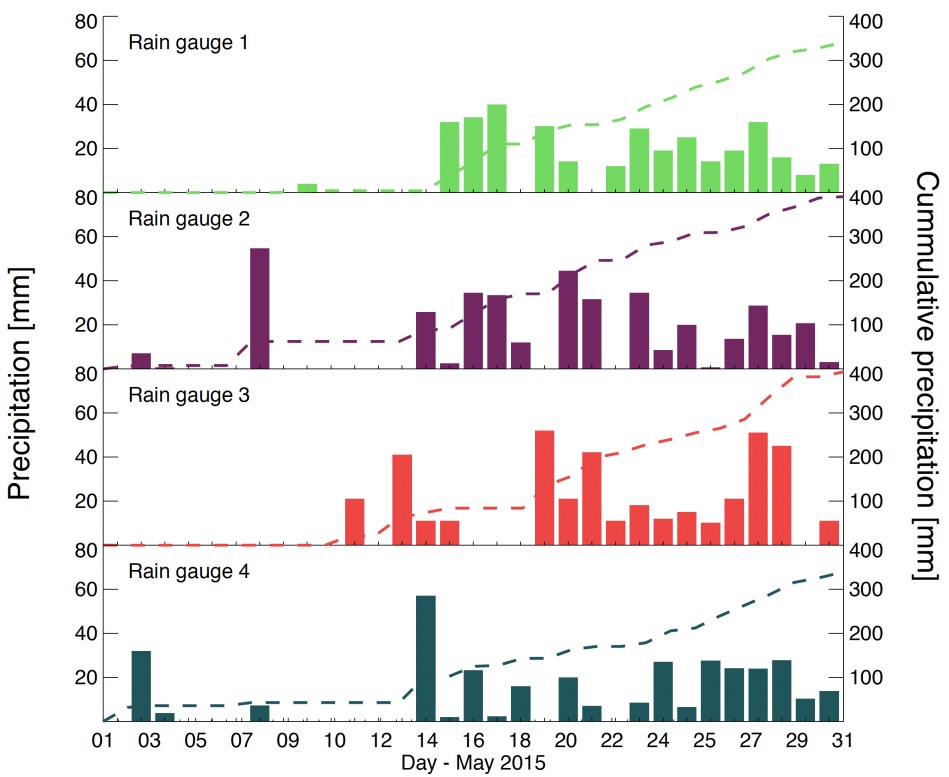

**Figure 3.** Daily precipitation during May 2015 near Salgar, recorded at the IDEAM in situ rain gauges shown in Figure 1. Bars correspond to daily cumulative precipitation and the dashed lines to the evolution of the cumulative precipitation during May 2015.

Similarly, a climatological assessment suggests that the monthly cumulative precipitation during May 2015, rather than being
anomalously high as a result of regional or global-scale forcing, it corresponds to magnitudes close to the May climatology in
the region. Figure 4a shows the annual cycle of precipitation and the multiannual average cumulative precipitation registered
at the in situ stations and the corresponding TRMM satellite record over La Liboriana basin. The Figure also shows the 5-year
average monthly rainfall, from 2014 to 2018, estimated using the QPE technique. Precipitation in the region exhibits a bimodal
pattern with peaks in the April-May and October-November periods modulated, from a climate perspective, by the migration
of the ITCZ (Poveda et al., 2005, 2006). According to in situ historical records, the annual precipitation in the region around
La Liboriana is between 2400 and 3050 mm, and during the peak months, precipitation reaches between 200 and 350 mm.
Precipitation data from TRMM shows that during May, the mean cumulative precipitation is 415 mm with a standard deviation
of 170 mm. La Liboriana flood occured during May, month climatologically corresponding to the highest precipitation in the
region. The QPE rainfall over La Liboriana basin appears to be higher than in nearby regions, and higher than the TRMM

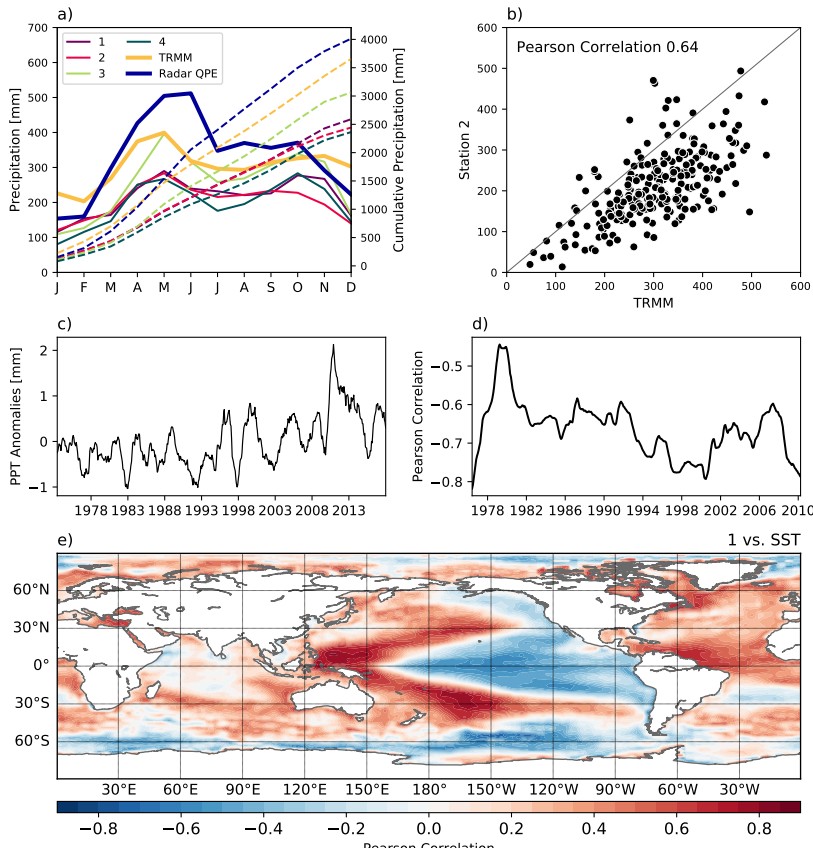

**Figure 4.** a) Mean annual cycle of precipitation near Salgar from IDEAM rain gauges and TRMM 3B42 product (thick yellow line) using data from 1998 to 2018. The panel also shows the 5-year average monthly QPE rainfall from 2014 to 2018. b) Scatter plot between monthly TRMM records and one of the in situ gauges, showing the Pearson correlation between both series (0.64). c) Precipitation anomalies over La Liboriana, after filtering all variability with periods equal to or shorter than 13 months. d) 10-year windowed moving correlation between the Multivariate ENSO Index (MEI) and in situ precipitation. e) Spatial distribution of the correlation between filtered SST and in situ precipitation.

precipitation record. Although the period is not the same, these differences are linked to the orographic rainfall enhancement in the upper part of the basin as it will be discussed in the following subsections.

The TRMM record reproduces well the annual cycle of precipitation in the region but the magnitude appears to be overestimated; however, it is important to note that in situ rainfall stations are not located in La Liboriana basin. We assess TRMM precipitation considering that in many vulnerable places, especially in developing countries, it is the only information available in near-real-time. Figure 4b shows the scatter plot between monthly TRMM records and one of the in situ gauges, and the Pearson correlation between both series. The correlation between TRMM and the other three in situ gauges is similar in magnitude as the one shown in Figure 4b. The evidence suggests that, while TRMM appears to overestimate the monthly rainfall, the

interannual variability is represented satisfactorily, with a correlation of 0.64. Precipitation anomalies over La Liboriana, after filtering all variability with periods equal to or shorter than 13 months, exhibits an important interannual variability, with a pe-
riod between 3 and 7 years (Figure 4c). Figures 4d and e present evidence linking this variability directly to the spatiotemporal structure of El Niño-Southern Oscillation (ENSO).

   The spatial distribution of the correlation between SST and in situ precipitation (Figure 4e) closely resembles the ENSO horseshoe pattern in the Pacific Ocean (e.g. Diaz et al., 2001; Deser et al., 2010). The magnitude of the correlation in this region is considerably high in some areas, with absolute values over 0.8. The sign of the correlation indicates that high SSTs
in the central Pacific and the cold tongue are associated with negative precipitation anomalies over La Liboriana. The link is robust as it does not change sign over the years, but its strength has waxed and waned during the last 4 decades, as presented in Figure 4d showing the moving correlation between rainfall and the Multivariate ENSO Index (Wolter and Timlin, 2011). This change in correlation suggests other external forcing agents, such as the Atlantic Ocean (See Figure 4e), also play an important role in modulating interannual precipitation variability in the region. Figures 5a and 5b show the regional cumulative
precipitation from the TRMM 3B42 product and mean SST during May 2015, respectively, and Figures 5c and 5d show their anomalies relative to the long-term May conditions. The mean pattern shows warm ocean surface water north of the Equator, south of Central America between 80 and 110 W, and important precipitation over the ocean, also north of the equator, but not necessarily collocated with the warmer surface waters, but rather with the 27-29º C isotherms since global circulation plays a key role in determining the overall location of the convection hotspots (Hoyos and Webster, 2012). Over the ocean, the
maximum precipitation is located around 10N and 110W. There is also precipitation over Colombia, with larger values over the Amazon basin and, in particular, over the Colombian Pacific region. However, precipitation anomalies show that during May 2015, most of the country had lower precipitation than the long-term average.

   It is important to note that, while the climate conditions, in general, modulate the likelihood of extreme event occurrence (Haylock et al., 2006; Orlowsky and Seneviratne, 2012), La Liboriana flash flood happened during El Niño conditions and the
resulting negative precipitation anomalies over Colombia. From a risk assessment point of view, even under external forcing leading to less precipitation than the expected value, the likelihood of hazard occurrence is never zero: it is not sufficient to consider the monthly cumulative precipitation, but it is necessary to study the nature of the precipitation events adding up to the monthly values, particularly their moisture sources, the preferred trajectory of the events, and the dominant precipitation type in a particular region and season. Figure 6 shows the seasonal predominant precipitation tracks estimated using the radar
retrievals and the back trajectory analysis focusing on the parcels arriving to 500 hPa over La Liboriana basin for the December-to-February (DJF), March-to-May (MAM), Jun-to-August (JJA), and September-to-November (SON) trimesters. Following the Li et al. (1995) methodology described in the previous section, we analyzed all La Liboriana precipitation events from May 2014 to August 2018, averaging all the estimated precipitation velocity vectors obtained with the mentioned methodology over the basin. Considering the average speed and direction, each of the events is categorized into eight different classes depending
on the direction they come from (N, NE, E, SE, S, SW, W, and NW) plus one class, local formation, for the events with wind speeds close to zero. For the back trajectories, Figure 6 shows the evolution of the trajectories arriving to the 500 hPa pressure

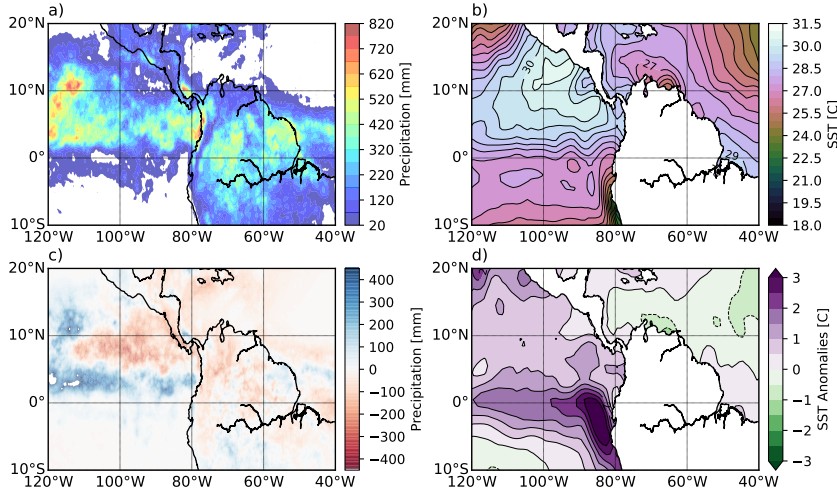

**Figure 5.** a) Regional cumulative precipitation from the TRMM 3B42 product (blue-to-red shading) and mean SST (green-to-white shading and black lines) during May 2015. b) Cumulative precipitation (red-to-white shading) and mean SST (green-to-pruple shading and black lines) anomalies relative to the long-term May conditions. Temperature contours are every 0.5ºC

level four days prior to the precipitation over the basin, indicating also the specific humidity of the parcels advancing towards the basin.

Results from both analyses show a predominant easterly flow and easterly propagation of the precipitation events towards La Liboriana basin. This trajectory is most characteristic during the JJA trimester, with almost 50% of the rainfall events over La Liboriana approaching from the southeast, and over 40% of the events from the east. During the other trimesters, there is also a predominant easterly flow towards the basin associated to the trade winds, but there is also important moisture influx from the Pacific Ocean, especially during SON, and a large percentage of local convective activity, mainly during DJF. The number of events leading to precipitation over La Liboriana with air masses approaching from the west is small, but the specific humidity is considerably larger than the easterly events. Considering the orientation of the basin and the overall topographical features, together with the east-west flow in the region, the likelihood of orographic precipitation enhancement is very high. During May 2015, the spatio-temporal evolution of the precipitation events followed the typical climatological pattern for the region (Figure 7a and b). It is important to note that moisture may reach the atmospheric column over La Liboriana at different levels. For this reason, Figure 7c shows the back trajectories for the events during May 2015 reaching the basin at 850 hPa. The May 18 (starting May 17) event (first convective core reaching the basin around 00:10 LT), moisture was advected from the east, while during May 17 (starting May 16), parcels reached La Liboriana from the west at 500 hPa, and from the southwest at 850 hPa, starting in the Pacific Ocean, near the Equator, off the coast of Peru, with higher specific humidity than during the 18th. This fact has two important practical implications: (i) the anomalous conditions leading to La Liboriana flash flood are not associated with marked changes in the predominant flow during May 2015 compared to climatology, and that (ii) the changes are more subtle than just considering the sign and magnitude of precipitation anomalies or distinct circulation patterns,

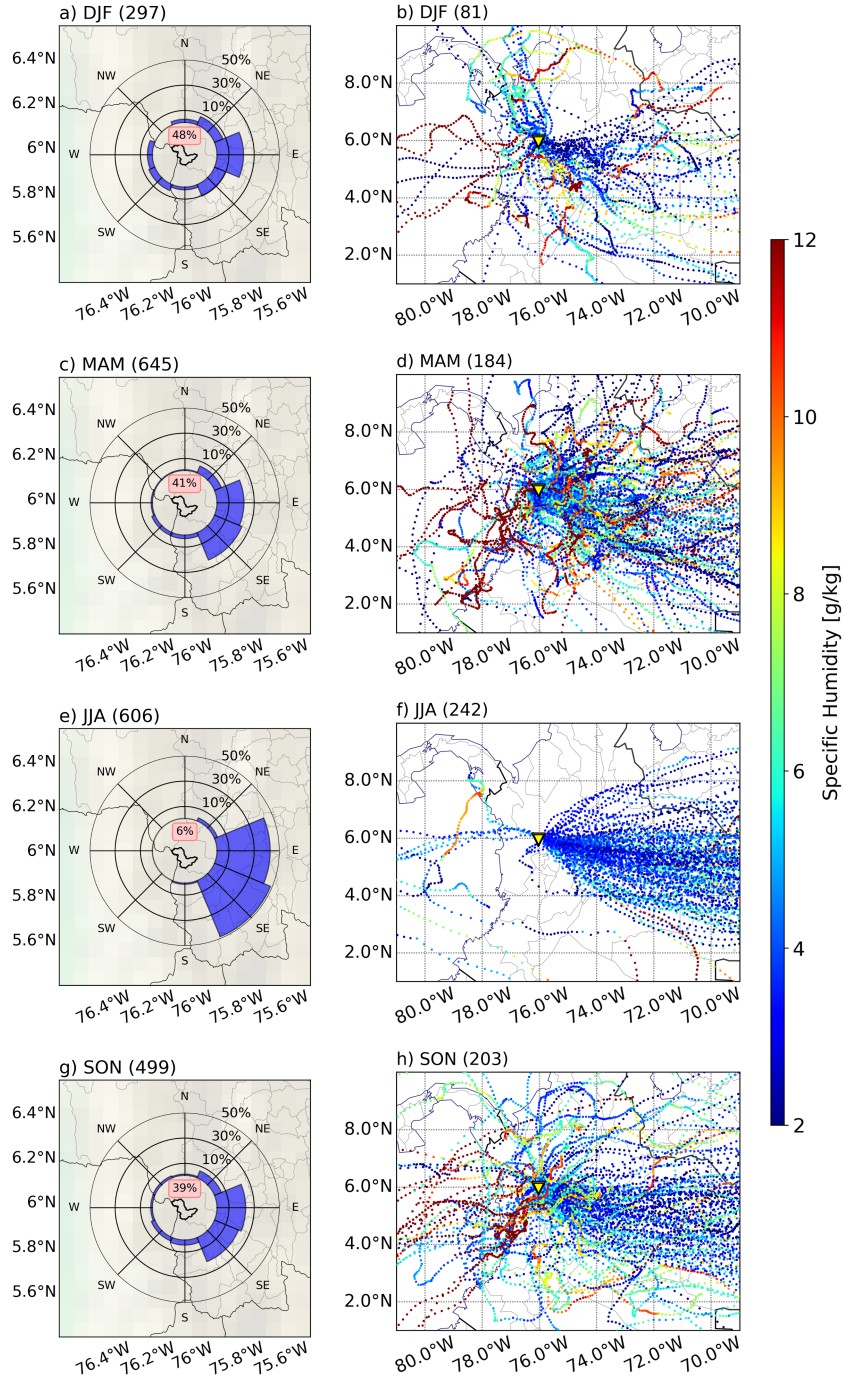

**Figure 6.** Seasonal predominant precipitation tracks estimated using radar retrievals (panels a, c, e, and g) and the back trajectory and specific humidity analysis arriving at 500 hPa over La Liboriana basin (panels b, d, f, and h) for the December-to-February (DJF), March-to-May (MAM), Jun-to-August (JJA), and September-to-November (SON) quarters, respectively. The percentage in the center of the diagrams in panels a, c, e, and g correspond to portion of the events categorized as of local formation. Each panel includes, in the title, the number of events considered in each case.

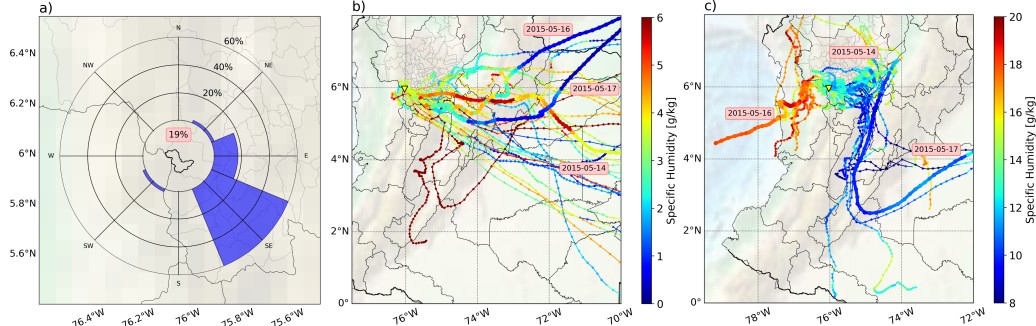

**Figure 7.** a) and b) Similar to Figure 6 for May 2015. c) Back trajectory and specific humidity analysis arriving at 850 hPa over La Liboriana basin.

thus implying that under near-average conditions La Liboriana and other similar basins on the Andes Cordillera, are subject to extreme hydrometeorological events, which translates into a challenge regarding risk management.

Figure 8 displays the average cumulative precipitation for La Liboriana basin for the 2013-2018 period, obtained using the described QPE technique. The precipitation is shown for the DJF, MAM, JJA, and SON trimesters, and it is discriminated in the portion associated with stratiform and convective precipitation using the Steiner et al. (1995) method. As shown in Figure 8, and coinciding with the TRMM product in magnitude and timing (see Figures 4a and b), the MAM trimester corresponds to the rainiest time of the year, and at the same time, it is the only season with a significant portion of both stratiform and convective cumulative precipitation. Conversely, during JJA and SON the precipitation in the region is mainly stratiform. This feature is relevant as it suggests that flash floods and rainfall-triggered landslides are more likely during the MAM quarter. The type of precipitation, and in particular its intensity, is key to understand flash flood and mass wasting events. As mentioned before, flash floods are more likely associated with intense convective precipitation events (e.g. Šálek et al., 2006; Llasat et al., 2016). During MAM, long-duration and low-intensity stratiform events precondition the basin overall soil moisture, recharging the gravitational and capillary storages prior to the occurrence of short-duration high-intensity convective events, with the potential of triggering flash floods and landslides. The previous assessment is validated by the analysis of the historical disaster records available at https://www.desinventar.org, for the sub-region of the Department of Antioquia including the municipality of Salgar (the southwest region). The number of flash floods and torrential flows in the sub-region, from 1922 to 2019, are 6 during DEF, 25 during MAM, 13 during JJA, and 10 during SON.

## 3.2 Synoptic and meteorological conditions leading to the flash flood

### 3.2.1 Synoptic conditions

A series of intense storms occurred days and hours preceding the Salgar disaster. Figures 9a, c, and e present the spatial distribution of the 700 hPa geopotential height and the corresponding wind field for the closest time to the occurrence of the flooding in Salgar (01:00 LT of May 18, 2015), and the prior two precipitation events on May 15 and 17, at the same time. The

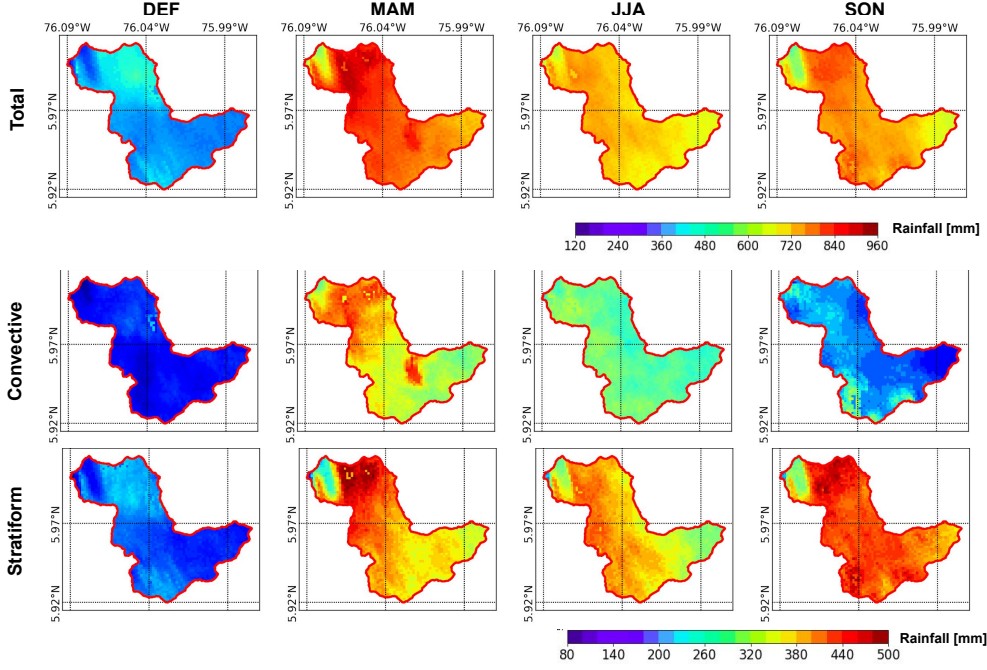

**Figure 8.** Precipitation climatology for La Liboriana basin for the 2013-2018 period using the described QPE technique. The precipitation climatology is presented for the DJF, MAM, JJA, and SON trimesters, and it is discriminated in the portion associated with stratiform and convective precipitation using the Steiner et al. (1995) method.

event during the 18th (Figure 9e) exhibits a predominant wind field from the southeast favoring air masses to reach the eastern slopes of the topographical ridges of the Andes Cordillera. This wind flow was favored by a low-pressure system that was
315 located over the Pacific coast of Colombia and Panama, and likely by the predominant easterly upper-level wind (not shown). Figures 9b, d, and f show brightness temperature (Tb) from the GOES-13 satellite for the same time as the wind field and the geopotential heights. Figure 9f shows an area of relatively low values of Tb (e.g., Tb<200 K) associated with a broad deep cloud system located in the northeast region of the Department of Antioquia, and a small cloud system in the southwest that is likely related to the storms that occurred in the day of the disaster in Salgar. The small footprint of these storms indicate the
320 precipitating elements that produced the observed rainfall correspond to early stages of localized intense convective storms. Later (not shown here), GOES imagery shows growth in the area of the cloud system, as the hydrometeors inside the cloud were lofted to the upper troposphere likely producing significant ice concentration, reducing the Tb measurements.

Twenty four hours before the flash flood (Figures 9c and d), the precipitation event was associated with a low-pressure center near the Pacific coast of Colombia, showing convergence in the region over the western slopes of the Andes in the Pacific region
of Colombia, affecting indirectly the municipality of Salgar. This feature is a characteristic pattern of the synoptic conditions associated with the occurrence of broad MCSs over this region (Houze et al., 2015; Zuluaga and Houze, 2015). In fact, the

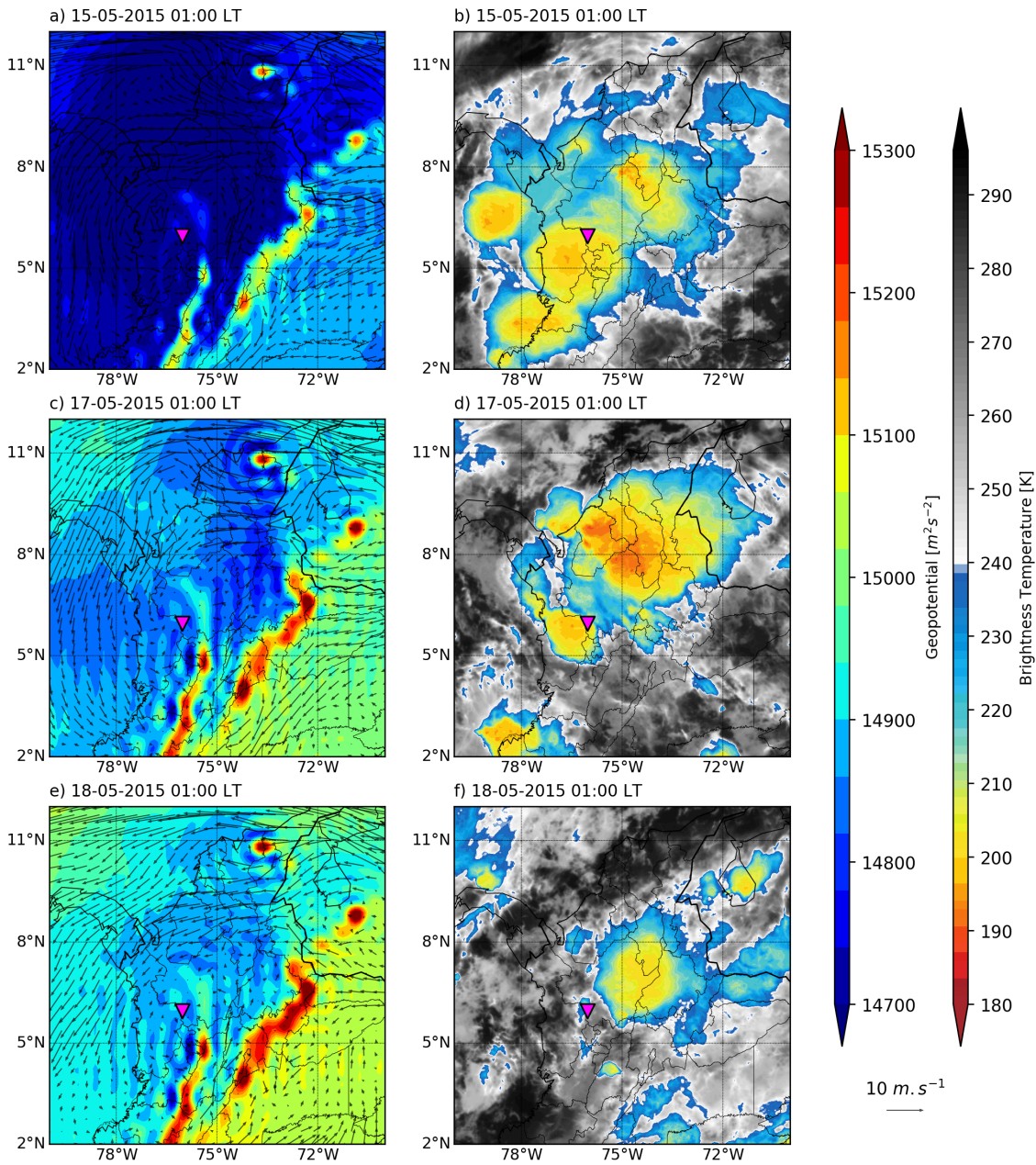

**Figure 9.** Panels a), c), and e) show the ERA Interim geopotential height at 700 hPa (filled contours) and horizontal wind (vectors) for May 15, 2015, May 17, 2015, and May 18, 2015, all at 01:00 LT, respectively. Panels b), d), and f) show the GOES-13 brightness temperature for the same times as in panels a), c), and e).

cloud system in Figure 9d displays a considerable growth over the western region of Colombia, favoring humidity convergence and the generation of storms over the Liboriana basin and the town of Salgar. In addition to the May 18 and 17 events, there was a prior important precipitation event during May 15 synoptically similar to the event during the 18th, but more intense (Figures 9a and b). While the overall geopotential height was lower in the entire region, there was a marked low-pressure system that was located over the Pacific coast of Colombia, the Department of Antioquia and Panama, favoring the southeasterly flow towards the western slopes of the Andes Cordillera, and even cyclonic anomalies of relative vorticity over Salgar. These conditions generated the growth of extensive cloud systems over the region with significant stratiform and convective rainfall.

### 3.2.2 QPE and satellite precipitation

Figure 10 shows the cumulative precipitation during the duration of the three events in Figure 9, obtained using the radar QPE technique. The Figure shows the spatial distribution of precipitation for a 120 km radius area centered at the radar site, and for La Liboriana basin. Figures 10a and b show the cumulative precipitation during the 25 hours between May 14, 17:00 LT and May 15, 18:00 LT (period 1), Figures 10c and d the cumulative precipitation during the 13 hours between May 16, 21:00 LT and May 17, 10:00 LT (period 2), and Figures 10e and f the precipitation during 8 hours between May 17, 22:00 LT and May 18, 06:00 LT (period 3). During period 1, the northern and the western municipalities of the Department of Antioquia were under heavy precipitation, with rainfall accumulation over 150 mm in various regions including La Liboriana basin. An interesting characteristic of period 1 is that there were large rainfall accumulations mainly in the higher elevations and steeper slopes of the Liboriana basin (Figure 10b). Cumulative precipitation during this period was over 110 mm in the upper half of the basin, making this event the largest of the three in terms of total rainfall. Before this event, since May 1, 2015, rainfall over La Liboriana was less than 80 mm, well below climatological values. Regardless of its magnitude, the precipitation during period 1 did not cause flash flooding in Salgar, partly because of its long-duration and, in average, relatively low-intensities (despite having high intensity spells), but more importantly because the gravitational storage in the basin's soils was low due to the below average rainfall between the first and the 13th of May (Velásquez et al., 2018).

Figure 11 shows the 5-minute time series of convective and stratiform rainfall over the La Liboriana basin during the period of maximum accumulation of each of the three events, as well as the evolution of the basin's area covered by either stratiform or convective precipitation. Overall, the Figure 11 shows that most of the rain accumulation over La Liboriana basin days before the time of the disaster was due to convective precipitation, with very intense spells. Figure 11a shows that a large part of the rainfall during the May 15 event was produced by convective precipitating clouds, and that during the most intense phase, lasting about 120 minutes, the area of the basin with convective precipitation was about the same as the stratiform-covered region. By the end of the event, although the accumulation is low, there is a predominance of stratiform rainfall over the basin.

The second of the three events started during the night of May 16, and the average cumulative precipitation during period 2, in the basin, was around 45-50 mm (Figures 10c and d). The spatial distribution of rainfall during period 2 is the most homogeneous among the three events, typical of stratiform-dominated events covering the entire basin as observed in Figure 11d, and that was part of a broad cloud system that was covering a significant portion of western Colombia. By the end of

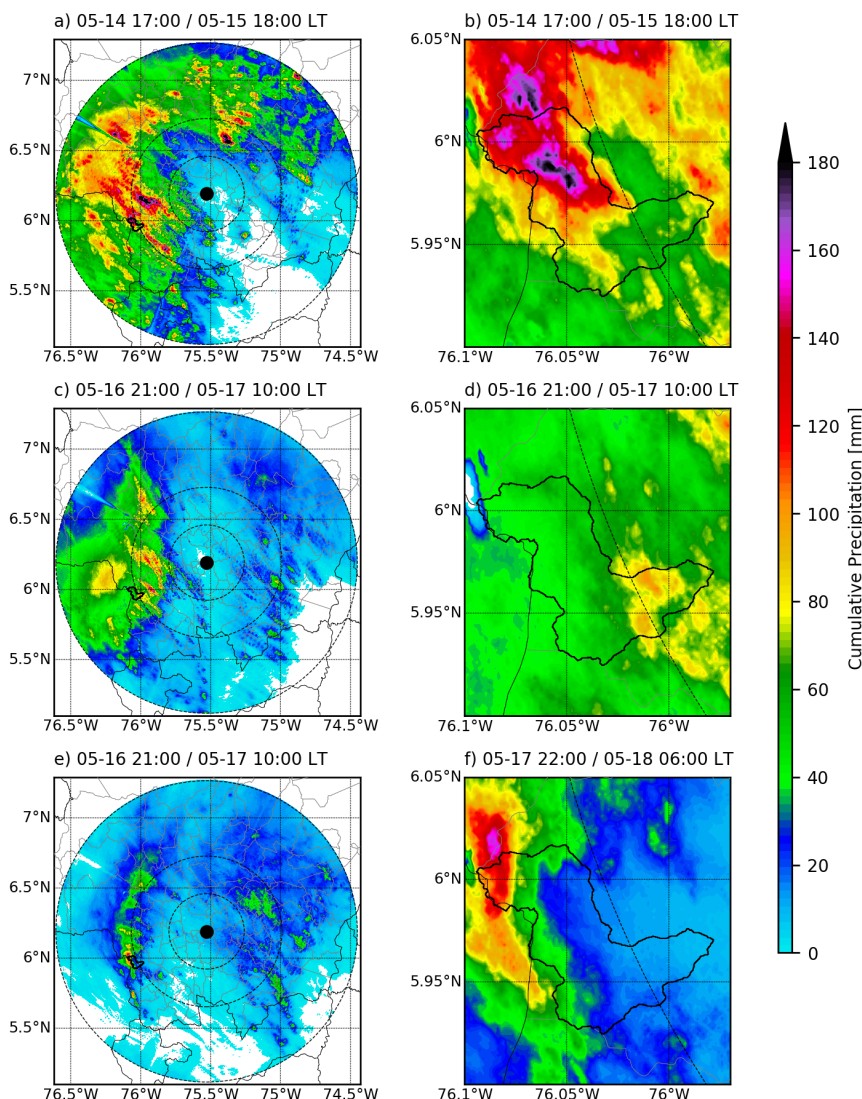

**Figure 10.** Cumulative precipitation for the three events prior to La Liboriana flash flood obtained using the radar QPE technique. The panels show the spatial distribution of precipitation for a 120 km radius area centered at the radar site, and for La Liboriana basin. a) and b) Cumulative precipitation during the 25 hours between May 14, 17:00 LT and May 15, 18:00 LT, c) and d) cumulative precipitation during the 13 hours between May 16, 21:00 LT and May 17, 10:00 LT, and e) and f) precipitation between May 17, 22:00 LT and May 18, 06:00 LT.

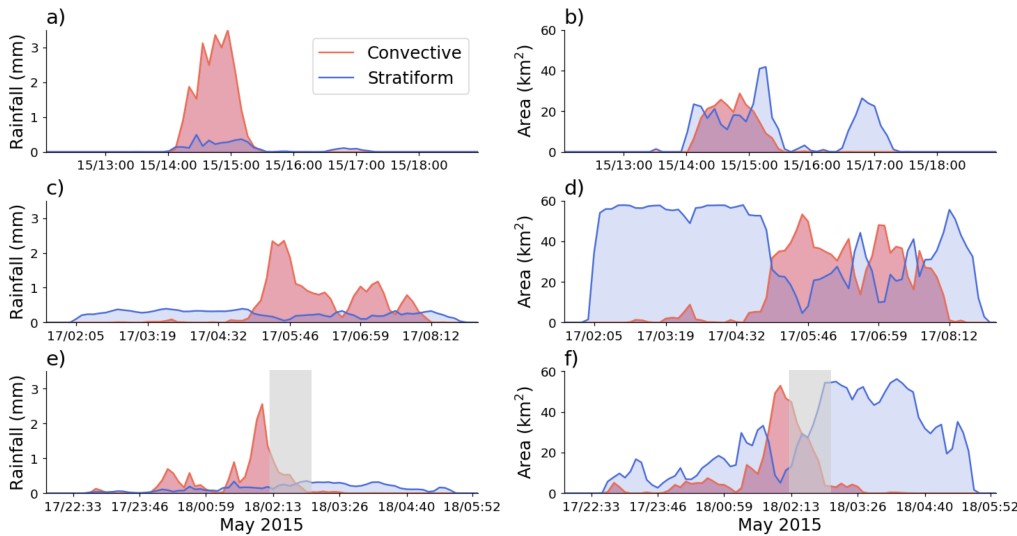

**Figure 11.** 5-minute time series of convective and stratiform rainfall over the La Liboriana basin during the period of maximum accumulation of each of the three events in Figure 10, as well as the evolution of the area of the basin covered by either stratiform or convective precipitation. a) and b) correspond to the May 14-15 event, c) and d) to the May 17 event, and e) and f) to the May 17-18 triggering event. The gray shadow in e) and f) corresponds to the approximate timing of the flash flood event in different parts of the basin according to local authorities and the community (from 02:10 to 02:40 LT).

period 2, Figure 11d shows the presence of three short-lived convective cells in the basin covering an important fraction of the basin.

Figure 12 shows weather radar horizontal reflectivity ($Zh$) retrievals depicting key moments of the evolution of this event (period 2) over the entire 120 km radius area, and over La Liboriana basin. The black dots correspond to cloud-to-ground lightning. Figures 12a to f show a region of stratiform rain that moved northeastward to La Liboriana from the Department of Chocó (the rainiest region in Colombia and one of the wettest of the world (e.g. Poveda and Mesa, 2000)), across the western branch of the Colombian Andes, generating close to uniform precipitation over the basin for a prolonged period. Eventually, convective cores that formed over the western hills of the *Cordillera Occidental* crossed La Liboriana generating intense precipitation that concentrated in the lower half of the basin and near the outlet (Figures 12i to l), where the hills are considerably flatter than in the upper half of the basin.

Rainfall during periods 1 and 2 increased the overall soil moisture in the basin, likely decreasing the magnitude of the infiltration rates (Penna et al., 2011; Zehe et al., 2010), hence increasing runoff and the likelihood of flash floods occurrence (Wagner et al., 1999; Penna et al., 2011; Tramblay et al., 2012). Twenty hours after the second event, a third event characterized by training convection (deep convective cells organized such that they move repeatedly over the same area as described by Doswell et al. (1996)) moved towards La Liboriana. Figures 10e and f show a considerably high cumulative precipitation over the highest elevation of the basin, towards *Cerro Plateado*, with values over 150 mm. The average precipitation over the basin

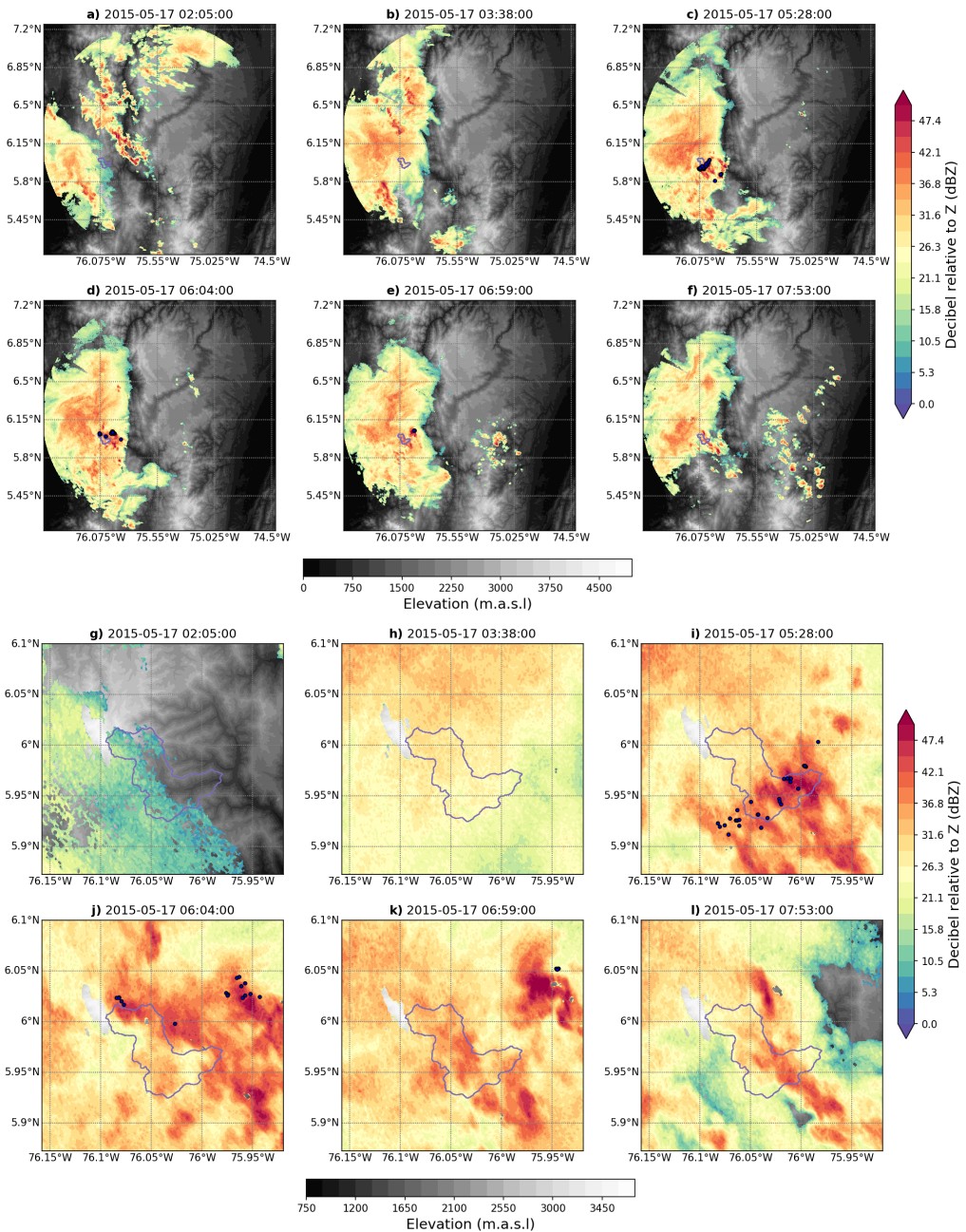

**Figure 12.** Quasi-horizontal reflectivity fields from the C-Band radar (1° antenna tilt) showing the evolution of the May 17 precipitation event (period 2) for a) to f) the region determined by the 120 km radius circular area centered at the radar site, and g) to j) for the region around La Liboriana basin at for the dates specified in each panel. The black dots correspond to cloud-to-ground lightning.

is between 35 and 40 mm. The evolution of the $Zh$ retrievals shows that the third event was composed of multiple convective storms, propagating in the southeast-northwest direction towards the upper part of the basin (Figures 13g to l), embedded in the convective envelope seen in GOES-13 Tb imagery (Figure 9f), and in the same direction as suggested by ERA-Interim low-level winds. In particular, there were two successive extreme convective cores generating rainfall with maximum intensity near *Cerro Plateado*, and ultimately triggering the flash flood. The first convective core entered the basin after 23:00 LT on the 17th of May (Figure 11e), intensifying towards the highest hills between 00:30 and 01:00 LT on the 18th (Figures 13i to j). At this point, $Zh$ values reached over 50 dBZ. Then, another intense convective core appeared around 02:00 LT, generating precipitation in the same places as the previous core and lasting for about 30 minutes (Figure 13k). The precipitating cloud associated with the second core had an elongated dimension of about 15 km by 5 km and was aligned SE-NW taking a similar shape as the basin. The flooding was reported in the urban area of Salgar about 30 to 45 minutes after the occurrence of this intense event. The convective cores were immersed in a system of storms that reached the mesoscale (around 70 km of length), and were moving across the Department of Antioquia with a SE-NW direction. These events can be categorized as almost purely convective (Figure 11e) with $Zh$ reaching over 40 dBZ. It is relevant to note that the cumulative convective rainfall reported during the day of the disaster was not the highest among the three events, and its intensity was not exceptionally higher.

Figure 14 shows the vertical cross-sections of $Zh$, radial velocity ($Vr$), differential reflectivity ($Zdr$), and polarimetric correlation coefficient ($hv$), across one of the convective cells that were occurring north of Salgar around the time of the disaster (02:00 LT, May 18, 2015) and was also immerse in the system of the storms. An intense, purely convective storm can be observed reaching up to 14.5 km (Figure 14a), with the characteristic divergent pattern in the upper levels of the radial velocity field (Figure 14b). The intensity of the storm is associated with a core of high $Zh$ values over a 7 km region (horizontal extension). Figure 14c shows values of $Zdr$ greater than 1, indicative of non-spherical particles likely composed of liquid droplets, associated with the high $Zh$ values between 4 and 6 km height, and also aloft, over 10 km height towards the western side of the storm, where ice and snow particle formation is likely occurring. All of this occurs in a region with high particle homogeneity, regarding hydrometeor shape, characterized by high $hv$ values (Figure 14d). The shape and values of the polarimetric variables observed in Figure 14 are highly indicative of the composition of the associated storms, showing mostly liquid droplets, and with particle microphysics distributed homogeneously over the affected region. The assessment of the vertical structure of the convective cores is also relevant to evaluate their intensification potential: in cases like the one presented in Figure 14, the depth of the convective core, and the overall structure of the system suggest a high potential for intense rainfall at the surface.

### 3.2.3 Orographic enhancement

The synoptic and meteorological evidence of 2015 La Liboriana flash flood and debris flow suggests an important role of orography in the spatio-temporal evolution of the triggering storms. Figures 15a to c show Hovmöller diagrams (longitude-time cross-section) of $Zh$ at 6ºN for different radar scanning tilts (1, 2 and 4º) relative to the topography of the region as shown in Figure 15d. The Hovmöller diagrams summarize the evolution of the flash-flood triggering event during May 18, 2015.

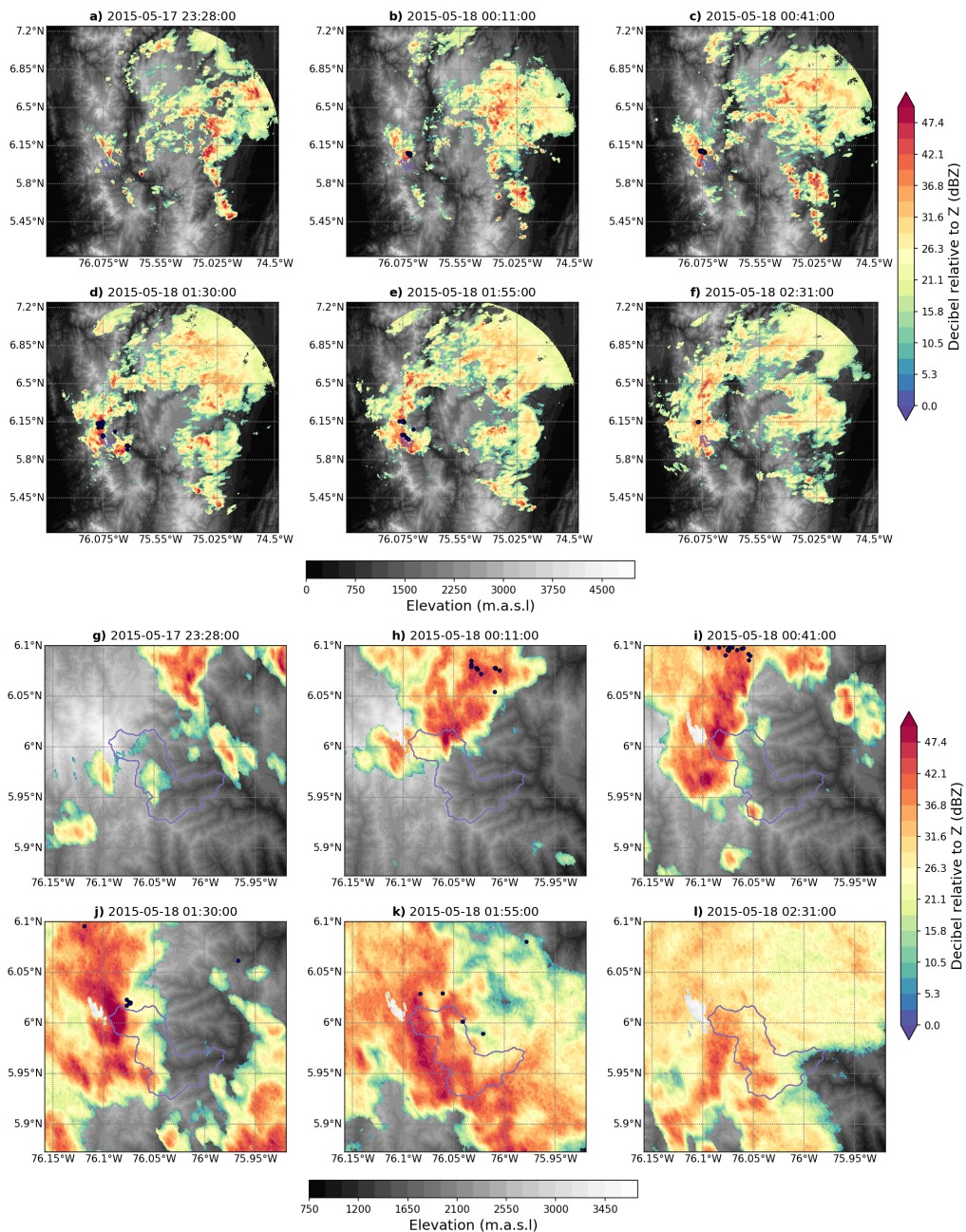

**Figure 13.** Similar to 12 for the flash flood triggering event (period 3).

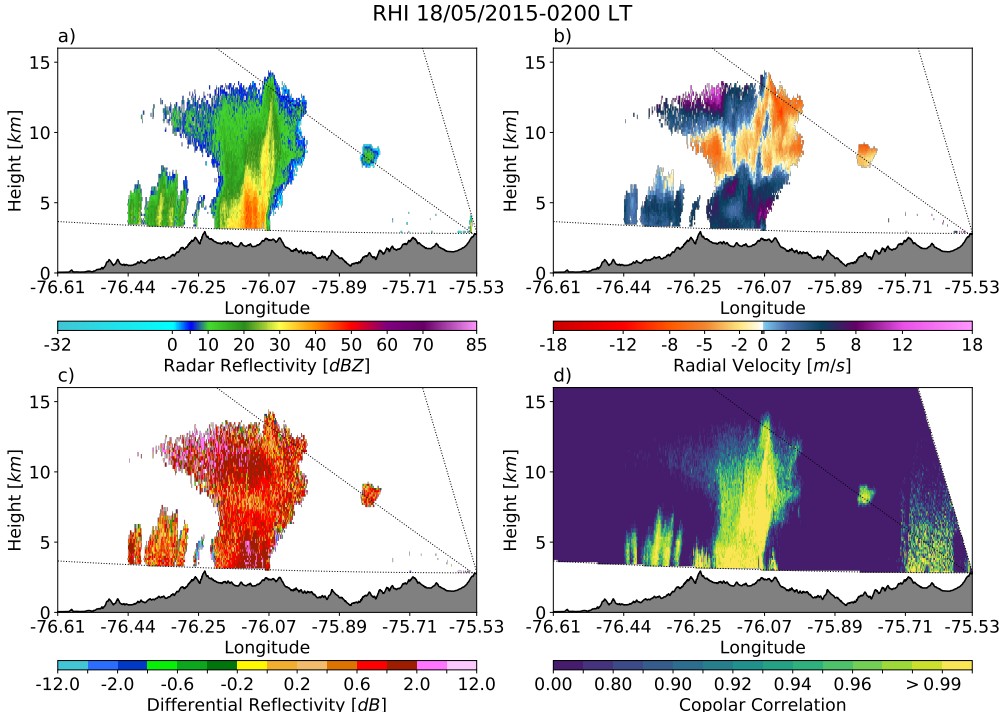

**Figure 14.** Vertical cross sections across the east-west direction (270°) of the reflectivity field shown in Fig. 1., showing a) horizontal reflectivity, b) radial velocity, c) differential reflectivity, and d) polarimetric correlation coefficient. Notice the maximum altitude of the reflectivity field is about 15 km and the base of the scans are about 3.2 km.

Figure 15e shows the temporal evolution of $Zh$, for the different tilts, along the black line in Figures 15a to c. Figure 15e also shows the corresponding terrain elevation below the propagating precipitating cloud. Overall, the Figure shows an event of medium intensities (20-30 dBZ) approaching the upper part of the basin from the east. As the convective cores approached the steepest topography, above 3000 m.a.s.l., the intensities increased significantly, with cores exceeding 40dBZ. At around 01:30 LT, a precipitation system with high intensities approached *Cerro Plateado* joining the first core, generating a high-intensity nucleus that extended latitudinally from 5.93°N to 6.06°N (see Figure 13). These intense cores persisted close to the steepest and highest terrain for about 2 hours when the system started to spread around, covering the basin with precipitating clouds of varying intensities (Figures 13k and i). The intensification of the cores as they approached the topographic obstacle is evident in the three radar scanning tilts (Figures 15a to c). Subsequently, the system began to dissipate as it migrated out of the basin in a northwestern direction. According to Figure 15e, it is possible to identify that the approaching systems increased their intensity by at least 30dBZ over the topographic barrier, highlighting the importance of the steep orography of the basin in the intensification of the cloud system generating the disaster over the municipality of Salgar.

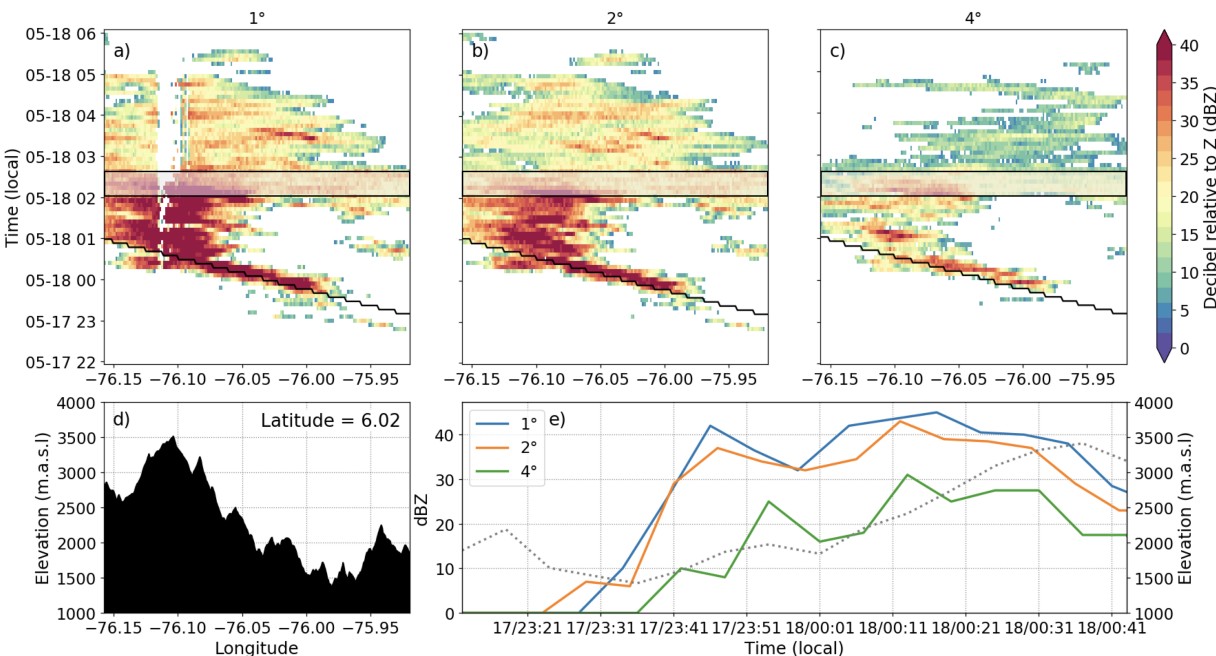

**Figure 15.** a) to c) Hovmöller diagrams (longitude-time cross-section) at 6ºN for different radar scanning tilts at 1, 2 and 4º, respectively. d) Topography of the region at 6ºN. e) Continuous lines show the temporal evolution of $Zh$ along the black line in a) to c), and the dashed line shows the corresponding terrain elevation below the precipitating cloud.

### 3.2.4   Evidence from satellite retrievals

Figure 16 shows the IMERG and TRMM 3B42 cumulative precipitation for the three events from May 14 to May 18, 2015, for the same regions and the same periods as in Figure 10. The regions shown, the 120 km radius area centered at the radar site ("radar scale") and the La Liboriana basin, allow assessing the usefulness of both satellite products in risk management. At the radar scale, despite the apparent underestimation of peak values by the satellite retrievals, the overall structure of the rainfall distribution is captured skillfully by the satellite-based estimates, and in particular by the IMERG algorithm, which is able to capture the spatial structure of the cumulative precipitation on all three cases, including the location and areal extent of the events. On the other hand, at the "basin scale", none of the products capture the intense cores evident in the QPE technique. This does not necessarily preclude the use of satellite information in risk analysis, but it does limit its direct use in flash flood warning systems. Alternatively, given that at the radar scale satellite information appears skillful, the spatial distribution of precipitation from IMERG, as an example, should be used as a two-dimensional probability density function (or mass spatial function) combined with downscaling schemes such as a multifractal framework (e.g. Deidda, 2000; Tao and Barros, 2010), to generate probabilistic higher-resolution precipitation fields conserving the original mass in the coarse scale.

Figure 17 shows Hovmöller diagrams of IMERG precipitation and GOES-13 Tb, at 6.00535ºN, for the flash-flood triggering event during May 18, 2015. Figure 17a presents evidence that the IMERG dataset captured reasonably well the event approach-

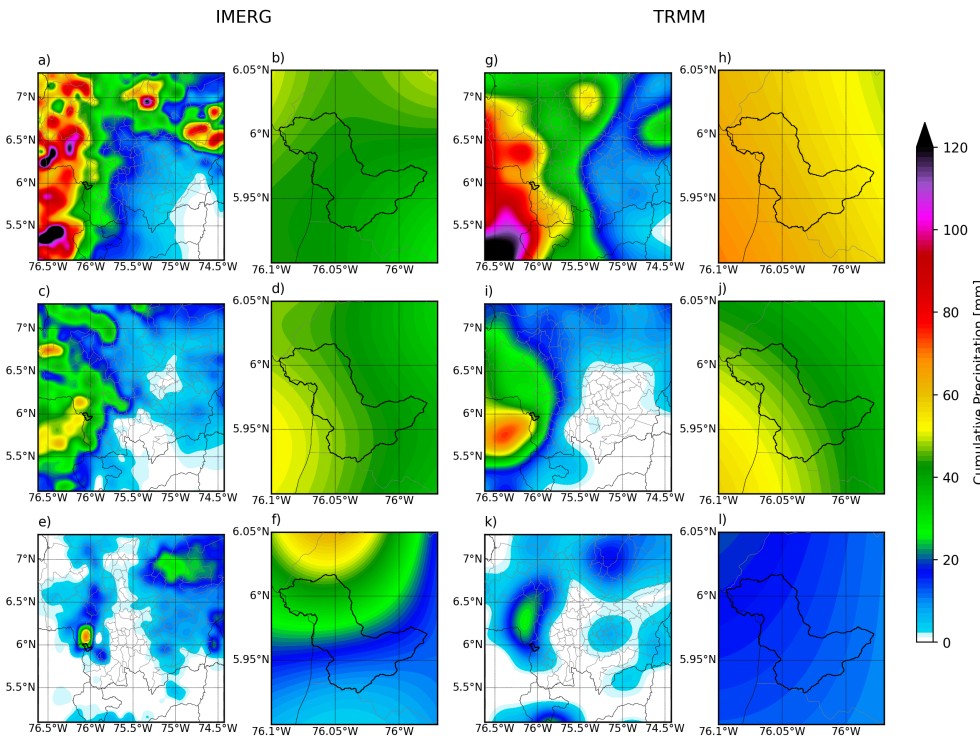

**Figure 16.** Same as Figure 10 for the IMERG (a, b, c, d, e, and f) and TRMM 3B42 (g, h, i, j, k, and l) products, correspondingly.

ing the basin from the east, and the main features of the orographic enhancement during the event, although with very coarse features. On the other hand, GOES-13 Tb, a proxy for cold cloud tops, does not clearly capture the observed orographic enhancement in cloudiness (Figure 17b) most likely due to its inherent synoptic nature: difficulties discerning deep clouds from thin-high clouds using the infrared-thermal channels.

### 3.3 First-order hydrometeorological processes

One of the goals of this work is to assess the likelihood of occurrence of extreme events similar to the one, or to the ones, triggering La Liboriana flash flood. In other words, it is important to evaluate whether or not the characteristics of the May 18, 2015 flood were exceptional, and ideally, their recurrence rate. In a traditional sense, it would be desirable to estimate a return period of the conditions that led to the la Liboriana flash flood. However, the length of the historical radar QPE record is not enough for a robust estimation of the return period. Considering this limitation, the previous analysis together with first-order hydrometeorological considerations allow us to conduct a preliminary assessment of the exceptionality of the precipitation conditions associated with the event. Analyses in the previous sections suggest that (i) the spatial structure of precipitation relative to the basin's main geomorphologic features, (ii) the occurrence of multiple precipitation events in a relatively short timespan (3-4 days), and (iii) the orographic enhancement of precipitation, all played a significant role in triggering the observed flash

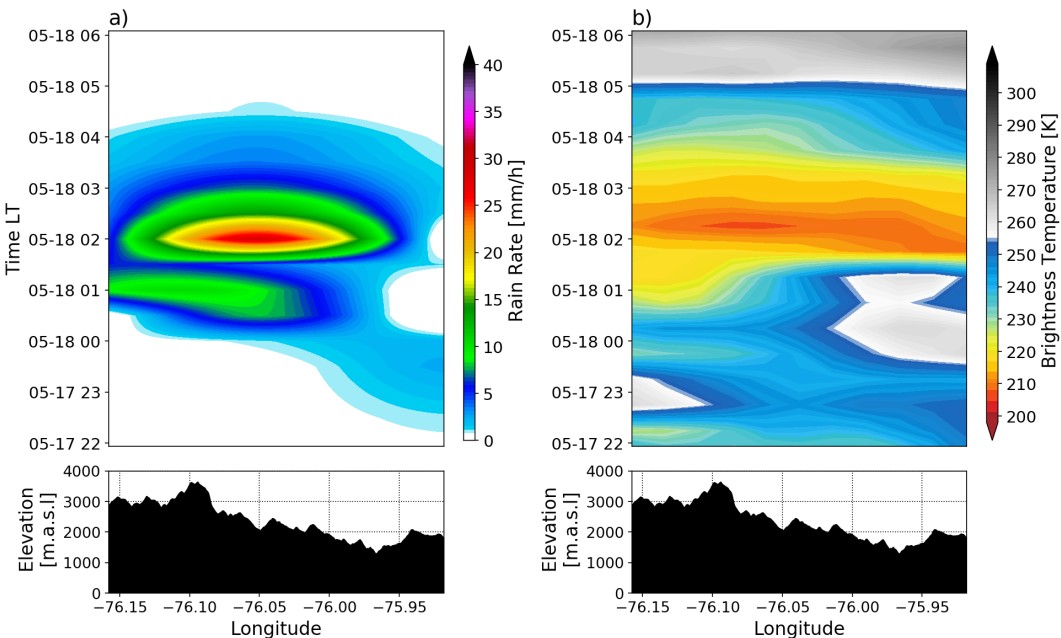

**Figure 17.** Longitude-time Hovmöller diagram at 6.00535 °N in the same time interval as before for a) IMERG interpolated rainfall rate, and b) GOES-13 brightness temperature. The terrain profile at 6.00535 °N is shown below the Hövmoller diagrams.

flood, suggesting that the traditional point rainfall return period estimation based on an intensity-duration-frequency curves (IDF), grossly misrepresents the observed hydrometeorological processes, even using areal transformations via reduction factors (e.g. Rodriguez-Iturbe and Mejía, 1974; Bacchi and Ranzi, 1996; Sivapalan and Blöschl, 1998; Veneziano and Langousis, 2005; Barbero et al., 2014). In general, all approaches considering as a basis a spatially random distribution of precipitation
over a specific basin would not represent properly the observations.

Figure 18 presents, from a to d, the spatial distribution of the 75th, 90th, 99th, and 99.9th hourly cumulative precipitation percentiles, respectively, for La Liboriana basin. The different percentiles are estimated considering the entire radar QPE record (2014-2018). The Figure shows, clearly, a preferential spatial distribution of the extreme rainfall over the basin, with higher values towards the upper basin. The peak hourly values in the upper basin, where the slopes are higher, change from
approximately 35 mm for the 90th percentile to 125 mm for the 99.9th percentile. The spatial distribution of the different percentiles is most likely due to the orographic intensification observed in the region. Figure 18e shows the hourly cumulative precipitation during the time of the flash flood, from 02:00 to 03:00 LT during May 18, 2015, including evidence of the intensification towards the upper basin, similar to the spatial structure of the different percentiles shown in Figures 18a to d. During the hour of the event, the cumulative precipitation in the upper part of the basin is in fact over the 95th percentile, a very
small percentage above the 99th percentile, but no region in the basin was above the historically observed 99.9th percentile (see Figures 18f): 87% of the basin was over the 90th percentile and 39% above the 95th percentile. The bivariate histogram in Figure 18g is evidence of the conditional link between the highest hourly cumulative precipitation and the steepest slopes in

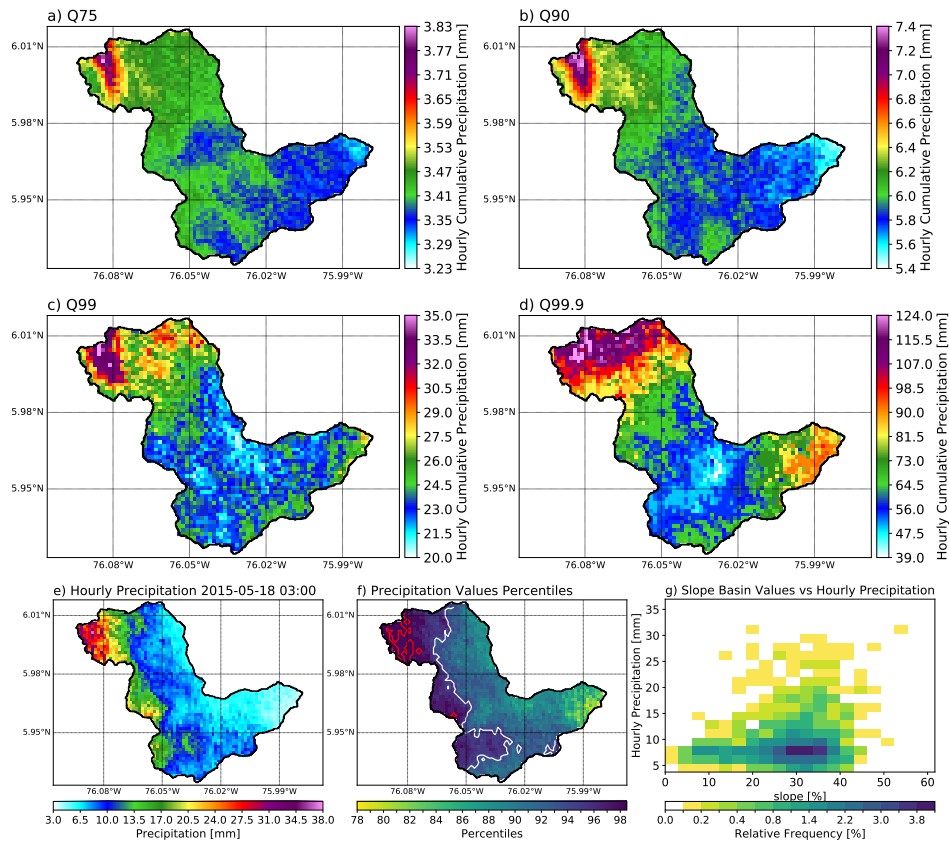

**Figure 18.** a) to d) Spatial distribution of the 75th, 90th, 99th, and 99.9th hourly cumulative precipitation percentiles, respectively, for La Liboriana basin. e) Hourly cumulative precipitation during the hour of the flash flood, from 02:00 to 03:00 LT during May 18, 2015, including evidence of the intensification towards the upper basin. f) Spatial distribution of the corresponding hourly percentile for the cumulative precipitation in e). The continuous white (red) line corresponds to the 95th (99th) percentile. g) Bivariate histogram of the hourly cumulative precipitation in e) and the slopes in the basin.

the basin, with the maximum cumulative values taking place only over regions with slopes between 20 and 40%, different to the lowest cumulative precipitation values that occur over the entire range of slopes. The converse relationship does not hold,
as the steepest slopes not always are associated with large values of cumulative precipitation.

Considering the evidence suggesting that the occurrence of the 2015 La Liboriana flash flood is linked to multiple precipitation events in a few days, or in other words, that no single precipitation event was exceptionally large to generate the extreme event, Figure 19 shows the assessment of the combined role of precedent rainfall modulating overall soil moisture, and the intense precipitation during the event, by estimating the bivariate histogram of 48-hour (and 96-hour) cumulative precipitation
and the hourly cumulative precipitation for the entire basin (Figure 19a -and c-) and for the upper basin (Figure 19b -and d-). The Figure also marks, for reference, the conditions associated with La Liboriana flash flood (black triangle in Figure 19).

Based on the historical record and the cumulative precipitation for the entire basin, the probability (long-term frequency) of having rainfall above the hourly La Liboriana average during the event is 3.05%, while for the upper basin is 0.65%. For the 48-hour cumulative precipitation, the probability of having larger values is about 1.93% for the entire basin, and 0.41% for the upper part. The bivariate probability for events above the one-hour and 48-hour cumulative precipitation associated with the flash flood is 0.16% for the entire basin and only 0.03% for the upper part. The latter probability, for the upper portion of the basin, corresponds to only two events for the entire record, both during the same week (16:00 LT May 15 and 03:00 LT May 18, during 2015). Aggregating the entire basin, eleven events show similar cumulative precipitation features in the historical record. The probabilities presented in this paragraph are not robust as they have been estimated based on a 5-year radar record; not enough for assessing extreme event recurrence as mentioned previously, however, the practical implications are important as the results suggest that La Liboriana event was, in fact, exceptional compared to all events in the 5-year radar record, but in particular, in the upper part of the basin, implying that for optimal risk management it is necessary to consider the spatial distribution of cumulative rainfall relative to the geomorphological features of the basin. In other words, while the individual event on May 18 was not exceptional, the climatological anomalies were negative-to-normal, and the synoptic patterns around the extreme event were similar to the expected ones for the region, the combination of high rainfall accumulation as a result of successive precipitation events over the basin, followed by a moderate extreme event is unique in the available observational record. The evidence for the La Liboriana basin also suggests that the 96-hour period is more appropriate to analyze the extreme event (Figures 19c and d). In this case, for the upper part of the basin, there is no other event in the historical record with one-hour and 96-hour cumulative values larger or equal to the Salgar event. An analysis of the historical disaster records available at https://www.desinventar.org for the municipality of Salgar, and the analysis and historical accounts presented in Polanco and Bedoya (2005) and Cardona-Duque (2018), show that there have been 11 extreme flash floods and torrential floods in the period from 1922 to 2019, five of them with fatalities. According to these reports, the largest so far corresponds to the event during Mary 18, 2015. The deadliest event before the La Liboriana disaster assessed in this study corresponds to a flash flood and torrential flows that occurred during June 1971, killing 45 persons.

## 3.4 WRF forecasts

The use of accurate and skillful numerical weather prediction models is arguably one of the most promising strategies to improve the lead times in flash flood forecasting schemes. In such a context, simulation and forecast skill refers, in general, to the ability of the model to capture the large-scale moisture advection features responsible for the heavy rainfall in a specific region (e.g. Younis et al., 2008; Gochis et al., 2015). In regions with complex terrains, such as Salgar, in addition to the moisture advection, limited-area models are required to represent the interaction of the flow with the local topography and all the processes leading to the observed orographic precipitation enhancement. In general, QPF precipitation forecasts of extreme events tend to underestimate the total rainfall amounts (Gochis et al., 2015); however, in cases where the spatial distribution of precipitation is accurately anticipated, model output statistics (MOS) techniques could be used to bias-correct the QPF for its use in flash flood likelihood assessment, providing valuable information for an optimal risk reduction.

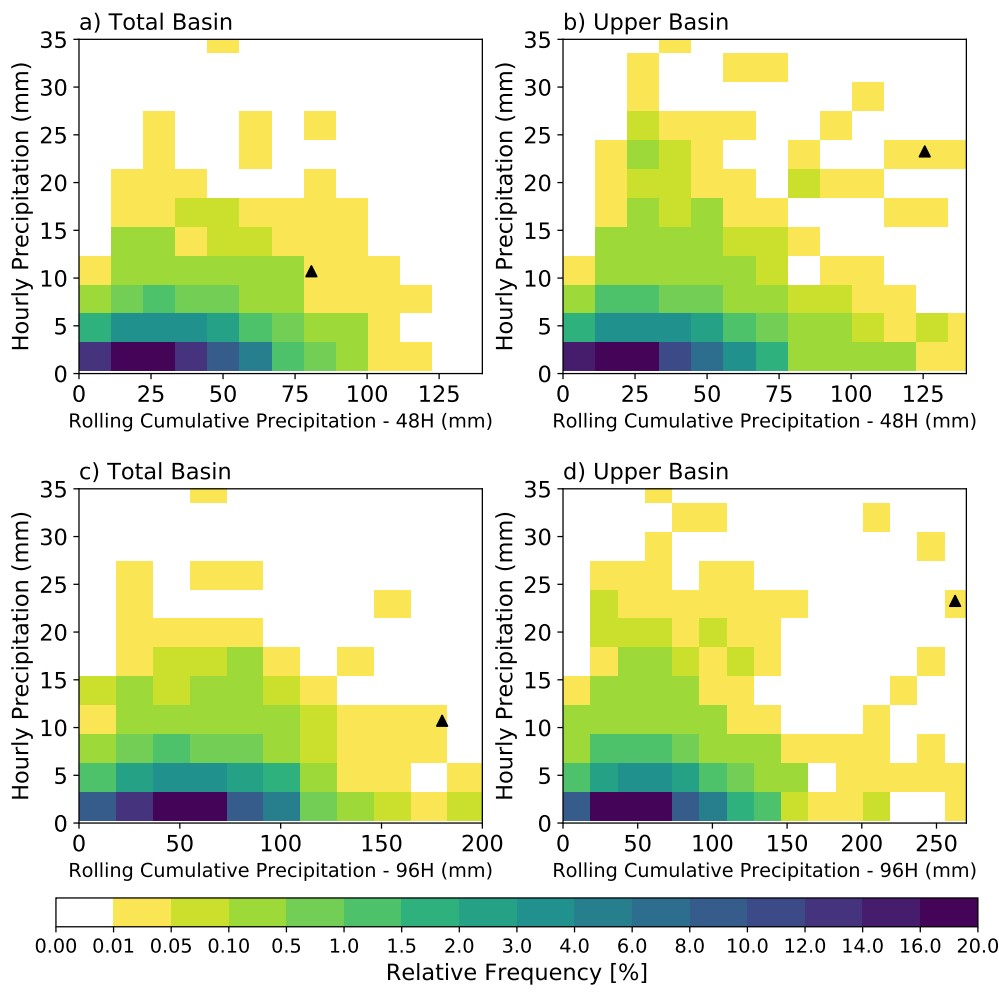

**Figure 19.** a) and b) Bivariate histogram of the 48-hour cumulative precipitation and the hourly cumulative precipitation for the entire basin, and for the upper basin, respectively. c) and d) Same as a) and b) for 96-hour cumulative precipitation. The black triangle corresponds to the La Liboriana flash flood event.

A brief evaluation of the limited-area operational weather forecasts issued by SIATA, suggests that the model simulates reasonably well the main processes leading to the observed orographic enhancement that took place prior to the La Liboriana flash flood. In the case of the SIATA operational forecasts, which tends to underestimate total rainfall amounts, we have found that the 90th percentile of precipitation in a given area represents better the observed average (or median) precipitation over the specific area, and we use it here as a simple quantile-to-quantile MOS technique. Figure 20 presents a summary of the assessment of the 120-hour operational forecasts issued daily, from May 14 to May 19, 2015. Different microphysics parameterizations were used operationally at the time, including the Single Moment 5-class, ETA (Ferrier, 1994) and the Lin (Lin et al., 1983) schemes. Here we present the evaluation of the model forecasts using the Lin et al. scheme given that, for the case of La Liboriana flash flood, the skill of the successive rainfall forecasts were considerably higher than using the other schemes.

Figure 20a shows the comparison between the hourly time series of spatially-integrated rainfall forecasts and the precipitation obtained using the radar QPE technique described previously. The observed time series is obtained by averaging the QPE over the Salgar municipality, and the simulated time series selecting the 90th percentile of the precipitation forecasts over the same region. In general, all the forecasts show heavy rainfall during the subsequent days matching the observations, especially the timing of the events. The May 14 forecast captures the timing of the series of all three events, but as lead-time is larger, it fails to capture the peak magnitude of the cumulative precipitation. Similar performance is observed for all the different forecasts, with the May 16 forecast being the most accurate capturing the rainfall event during May 17. Even though the event on May 18 was forecasted in all cases, the amount of precipitation was considerably less than the one observed. Figure 20b shows the total amount of precipitation accumulated from the start of the forecast to May 19. In general, despite using the 90th percentile, the forecasts underestimate the observed rainfall amounts. In the May 14 forecast, the total precipitation corresponds to 160 mm over the Salgar region, and the radar observations show 390 mm (May 17 and 18 events were considerably underestimated). Among all the forecasts shown in Figure 20a, the May 16 case shows a better agreement with observations as it captured the May 17 event, with a total accumulation of 120 mm compared to the observed 190 mm.

Figure 20c shows the spatial distribution of the May 14 1200 UTC 96-hour forecasted cumulative precipitation. While the forecast significantly underestimates the total amount of precipitation during the period in question (see Figure 10), the model captures considerably well the intensification in the upper part of the basin caused by orography. The operational WRF model, despite the relatively coarse horizontal resolution, generates an area of high precipitation towards the upper part of La Liboriana at *Cerro Plateado*, similar to the observations (Figures 10b, d and f). Figure 20d shows a scatter plot of hourly zonal wind at 2500 m.a.s.l. and cumulative precipitation at the surface over the location marked in Figures 20e and f by a red asterisk. Each data point in the scatter plot is color-coded using the magnitude of the relative humidity at the same location. Figure 20d suggests that, in the model, moist air is advected towards the topographic barrier, triggering precipitation. The Figure also suggests a conditional relationship between rainfall and the advection of moist air, in which precipitation in the region only occurs when the relative humidity of the air advected towards the upper basin is higher than 40%. Figures 20e and f show two snapshots of the model forecasts when there is orographic ascent of moist air, forced by the topographic barrier. In Figure 20e the orographic ascent is incipient, in the first stages of development, as a response to the easterly wind, towards the barrier,

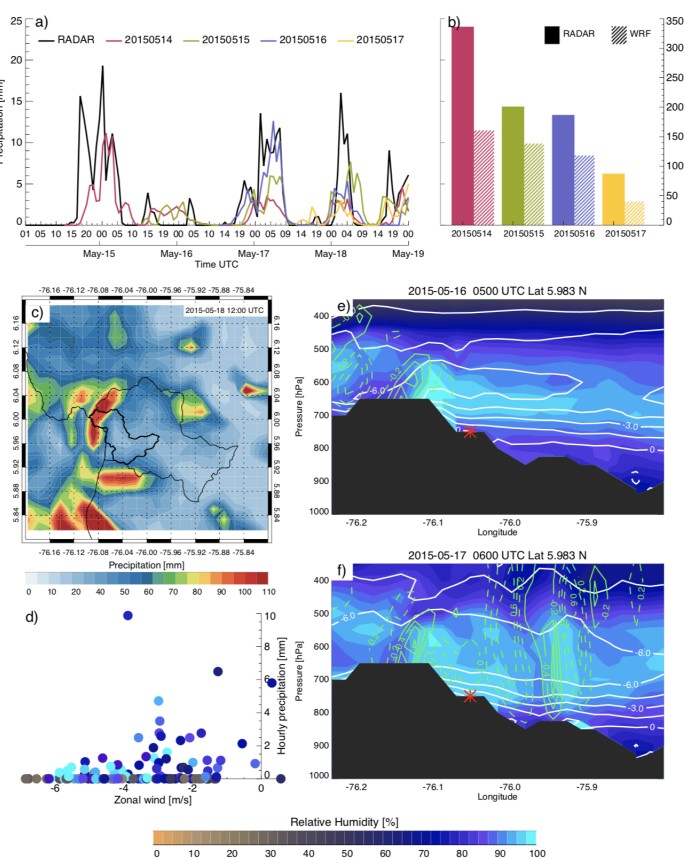

**Figure 20.** a) Hourly time series of precipitation over Salgar derived using the radar QPE technique and WRF forecasts. Each colored line corresponds to the model precipitation from different forecast dates, and the black line to rainfall estimations from radar QPE. b) Total amount of model precipitation over Salgar, accumulated from the start of each forecast to May 19. c) Spatial distribution of the May 14 1200 UTC 96-hour forecasted cumulative precipitation over the region of interest. d) Scatter plot of hourly zonal wind at 2500 m.a.s.l. and cumulative precipitation at the surface over the location marked in e) and f) (red asterisk). e) and f) Snapshots of the model forecasts showing a clear orographic ascent of moist air, forced by the topographic barrier. The colors represent relative humidity, the continuous white line represent easterly wind, and green continuous (dashed) lines represent subsidence (ascent). The black area represents the topography of the region. The relative humidity color table describes the colors in panels d), e) and f). The magnitude of the zonal and vertical winds shown in the contours are in ms$^{-1}$

.

generating forced ascent and moistening of the air near the barrier (around 76.1°W) and away towards the east at around 670 hPa. Figure 20f also shows another case of orographic lifting during a more mature stage, when the ascent of air occurs in a broader column, penetrating deeper into the upper troposphere. In order to evaluate the actual usefulness of WRF in flash flood likelihood assessment, it is important to discard the possibility that the WRF-simulated rainfall pattern associated with La Liboriana flash flood (Figure 20c) is an artifact of the model, overemphasizing the orographic enhancement of precipitation.

During the April-May 2015 rainy season, radar observations show a total of 57 rainfall events in addition to the ones leading to La Liboriana. Among all 57 events, 42 show a similar synoptic pattern. On the other hand, WRF simulations of the 42 events, from the daily operational SIATA weather forecasts, do not exhibit extreme orographic intensification (see Figure A.1 for the spatial pattern of the WRF-simulated cumulative precipitation for the top nine rainfall events our of the mentioned 42 cases). This suggests that the WRF simulation during the period of La Liboriana is not an artifact of the model, providing useful

information for risk management.

## 4    Discussion and conclusions

Flash floods are a recurrent hazard for many developing Latin American regions as a result of the interaction between copious rainfall and the complex mountainous terrain associated with the Andes Cordillera. Additionally, these regions often lack the timely and high-quality information needed to assess, in real-time, the threats to the vulnerable communities due to extreme

hydrometeorological events. The mitigation of the flash flood risk in a region of interest requires anticipation with different lead times. Long-term strategic actions require, at least, a process-based evaluation of flash flood potential in the region. On the other hand, the short-term tactical response ought to be based on real-time assessment of the observations, and simulation and prediction with useful lead times. Past extreme events become an opportunity to improve our anticipation capabilities to strive for the protection of life, so that extreme events like the one that occurred in May 2015 in La Liboriana basin do

not result in loss of life. The systematic assessment of the climatological aspects, meteorological conditions, and first-order hydrometeorological processes associated with La Liboriana flash flood allowed us to analyze the main triggering processes, highlighting critical lessons to improve local risk reduction strategies. Regarding the lead times, in the case of La Liboriana, an analysis of the lag between the peak discharge time relative to the maximum intensity for the events presented in Velásquez et al. (2018) suggests the minimum useful forecast lead time is approximately 1.5 hours. This lead time also matches the

estimates of the time of concentration of La Liboriana basin (between 1.3-1.7 hours) using different methodologies, including the Kirpich (1940) and Giandotti (as cited in (Fang et al., 2008)) equations.

The overall evidence suggests that, from a climatology point of view, (i) La Liboriana flash flood took place during a period with negative monthly precipitation anomalies associated with El Niño conditions, however, (ii) the long-term back trajectory and Lagrangian analyses show a predominant easterly flow and propagation of the precipitation events towards La

Liboriana basin, which, given the orographic opposition to the prevailing wind, stress the potential for orographic rainfall enhancement. That is the case of La Liboriana flash flood during 2015, characterized by moist easterly flow towards *Cerro Plateado*, permitting orographic rainfall enhancement. Also, (iii) during MAM, due to the considerable total precipitation

accumulation and the coexistence of long-duration and low-intensity stratiform events with short-duration and high-intensity convective events, the potential for triggering flash floods and landslides is higher than during other seasons. The climatological

assessment provides the first important lesson for risk management: In most Latin American countries there is a media and institutional bias to link all risk management planning to whether there are, or there could be El Niño or La Niña conditions during a certain period, and, while that is certainly important and an improvement compared to the previous decades, it should not be the sole focus, especially in regions like La Liboriana basin, where the interaction between the geomorphological features and the predominant lower troposphere flow could likely result in orographic intensification and the occurrence of

extreme events. Even in regions with scarce information, a combination of satellite rainfall retrievals (i.e., IMERG), detailed atmospheric reanalysis information (i.e., ERA 5), and the available high-resolution DEMs provide the means to pinpoint the most susceptible regions to prioritize the required long-term strategic actions. This analysis should allow the holistic detection of susceptible zones to flash flooding.

The analysis of the spatio-temporal configuration of precipitation, identifying the main hydrometeorological factors control-

590 ling the occurrence of extreme events and their likelihood of occurrence, is the foundation for the second risk management lesson. In the case of La Liboriana event, a series of intense storms associated with wind flow favored by a low-pressure system that was located over the Pacific coast of Colombia and Panama occurred days and hours preceding the disaster. The overall evidence of La Liboriana flash flood shows a definitive role of the multiple precipitation events in a relatively short period, and of their intensification as convective cores approached the steepest topography. There were three successive events generating

significant rainfall within La Liboriana basin: No single precipitation event was exceptionally large to generate the extreme event, but rather the combined role of precedent rainfall, and the extreme hourly precipitation triggered the event. The first two events, that increased the overall soil moisture in the basin, were followed by a third event characterized by training convective elements towards La Liboriana with significant orographic enhancement and a preferential spatial distribution of the most extreme rainfall towards the upper basin, ultimately triggering the flash flood. Most of the rain over La Liboriana basin days

before the disaster was of the convective type, with very intense spells, generating high rainfall accumulations with the highest hourly cumulative precipitation linked to the steepest slopes in the basin due to the orographic intensification in the region. Tactic risk management decisions would benefit from implementing, as a complement to flash-flood guidance tools, a real-time assessment of the bivariate distribution (and joint probability) of each ongoing event compared to the historical record, taking into account high one-hour moving cumulative rainfall conditioned to values of precedent $N$-hour moving cumulative precip-

itation. This analysis should be conducted for the entire basin as well as for critical areas associated with, for example, the steepest hills. The value of $N$ should be defined for each basin as a function of the general geomorphology and the historical records. From this point of view, the results suggest that the rainfall prior to the Salgar event was, in fact, exceptional compared to all the records, especially, in the upper part of the basin. The evidence also suggests that for La Liboriana basin, $N =$96-hour period is more appropriate to assess the likelihood of the occurrence of an extreme event. The analysis presented here could be

extended to other complex terrain basins with vulnerable communities, considering that it is likely that for each different basin there is a different optimal $N$-hours cumulative precipitation to be considered in the construction of the bivariate histograms by risk managers.

The present study also highlights the usefulness of long-term QPE records based on radar retrievals from high-resolution scans, in assessing the spatial structure of flash-flood triggering rainfall events. Low-resolution radar scans ( 1 km grid spacing) are useful from the meteorological standpoint, but high-resolution retrievals are required for hydrometeorological applications. Also, for information-scarce regions, the study suggests that IMERG captures the general spatial structure of the cumulative precipitation of extreme events, including their location and areal extent, even in cases where the complex topography leads to orographic enhancement. Even though IMERG does not capture the most intense convective cores evident in the radar retrievals, the spatial distribution of precipitation could be used as a starting point for a probabilistic rainfall downscaling scheme, useful for risk management applications. Lastly, WRF simulates reasonably well the main processes associated with the observed orographic enhancement of rainfall, capturing the intensification in the upper part of the basin as moist air is advected towards the topographic barrier. The latter fact suggests that forecasts from WRF are useful and should be taken into account in warning operations.

*Code and data availability.* The radar data and the WRF namelist and overall setup will be made available upon request.

*Author contributions.* CDH conceived the research with help from MDZ, LIC, and NV. CDH prepared the manuscript with contributions from all co-authors. LIC, LHM, and CDH conducted the climatological analysis. JSPC and OH performed the back-trajectory and the Lagrangian analysis. JS estimated the surface precipitation using Radar retrievals. NV conducted the stratiform-convective separation. MDZ helped with the synoptic analysis. SMLZ, JSPC, and JS performed the orographic analysis and the hydrometeorological assessment. LIC, MZ, and GGE performed the WRF runs and analyzed the forecast output.

*Competing interests.* The authors declare no competing interest.

*Acknowledgements.* This work was partly supported by SIATA (Sistema de Alerta Temprana de Medellín y el Valle de Aburrá) funds provided by Area Metropolitana del Valle de Aburrá (AMVA), Municipio de Medellín, Grupo EPM, and ISAGEN under the Research and Technology Contracts CD511, 2017. M. Zuluaga and L. Herrera were supported with resources from *Patrimonio Autónomo Fondo Nacional de Financiamiento para la Ciencia, La Tecnología y la Innovación, Francisco Jose de Caldas*, from COLCIENCIAS, under the contract 80740-128-2019 with Universidad Nacional de Colombia and AMVA. The authors would like to thank Professor Freddy Vinet, and the anonymous reviewer for their insightful comments.

## Appendix A: Figures

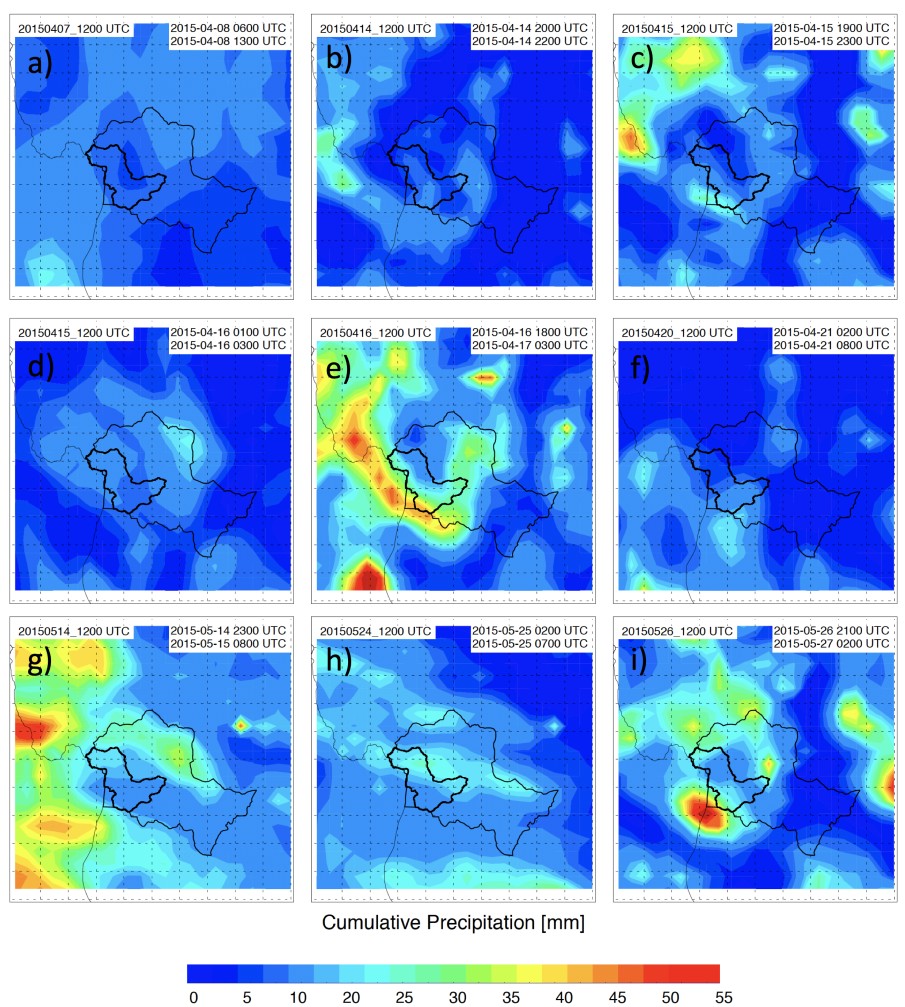

**Figure A.1.** Spatial distribution of WRF-forecasted cumulative precipitation (24 hours lead time) over the region of interest. for nine different rainfall events during the April-May 2015 rainy season: a) from 2015-04-08 0600 to 2015-04-08 1300 UTC, b) from 2015-04-14 2000 to 2015-04-14 2200 UTC, c) from 2015-04-15 1900 to 2015-04-15 2300 UTC, d) from 2015-04-16 0100 to 2015-04-16 0300 UTC, e) from 2015-04-16 1800 to 2015-04-17 0300 UTC, f) from 2015-04-21 0200 to 2015-04-21 0800 UTC, g) from 2015-05-14 2300 to 2015-05-15 0800 UTC, h) from 2015-05-25 0200 to 2015-05-25 0700 UTC, and i) from 2015-05-26 2100 to 2015-05-27 0200 UTC. While some of the cases exhibit orographic intensification, the panels show significant inter-event variability despite their similar synoptic flow (not shown), with no fixed orographic pattern indicating intensification at *Cerro Plateado*.

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
