# Peer review of "Meteorological Conditions Leading to the 2015 Salgar Flash Flood: Lessons for Vulnerable Regions in Tropical Complex Terrain"

_Natural Hazards and Earth System Sciences, 2019_

## Referee Comment (RC1) · Freddy Vinet (Referee) · 4 Aug 2019

The paper presents an overview of the meteorological conditions that triggered the flash flood in Salgar (Colombia) in May 2015. The authors put together a lot of information using radar imagery, satellite retrievals, atmospheric modelling and trajectories... The combination of different sources allowed a comprehensive overview of the case study.

The presentation of methods and sources is clear and relevant. Figures are useful and relevant. Captions are detailed and full of required information to understand the figure. Two figures would require some modifications in my opinion : - Figure 1a is small

and we are not able to locate the city of Salgar the main urban areas hit by the 18th may Flash floods in the frame of the catchment. - Figure 3 : rainfall amount usually are represented with bars instead of curves (curves are indeed reserved for cumulative precipitations - fig4f 4g , SST are not really visible. The presentation of results is also relevant showing that the floods results in a complex scenario associating previous huge rainfalls in the upstream basin and triggering rainfalls 3 days after although for this third episode, hourly precipitations was not so extreme. I suggest to replace hydrometeorological in the title by meteorological as there is no information in the paper on the hydrological characteristics. If authors want to keep the title "hydrometeorological " in the title they should introduce information on hydrology e.g. in the discussion comparing the occurrence of peak discharge and the highest hourly precipitations;

Discussion and conclusion The discussion is rather a conclusion. Some comparison with other cases would be welcome. Onanther hand, it is relevant to consider not only the rainfall of the 18th of May but also the cumulative rainfalls of the previous days but (p.28) the 96-hours cumulative precipitation forecast is suggested to be the best period to assess the potential occurrence of flash floods but this conclusion is valid only for this case study. In another case, the relevant period could be 100 hours or 108 hours etc. flood risk managers in Salgar may have at their disposal a running cumulative amount of rainfall over the basin for any period and above all, an updated assessment of soil moisture over the basin. But that is matter of hydrologists. One of the weaknesses of the paper is the shortness of the reference period to estimate the return period of the 2015 rainfall event. To conclude that the flash flood is the more intense over the last five years is not really interesting. Authors account for the radar QPE record over the period 2014-2018. We suppose that they have not any other long series at their disposal. The discussion could introduce some qualitative testimonies about previous flash floods (last century?) that would allow putting the 2015 event in perspective.

typing mistakes : Line 241 : would not the temperature be 27°-29° instead of 37° -39° ? Line 271 : It is important to note that the we use moisture may reach the atmospheric

column over La Liboriana at different levels. l. 272 : reaching l. 493 : Âń specific Âż p. 30 caption of figure 19 : "topography" and Âń The color table for relative humidity corresponds to panels.." l. 517 : suggests L. 528 : additionally
* * *

---

## Referee Comment (RC2) · Anonymous Referee #2 · 14 Sep 2019

The current manuscript deeply studies the different meteorological conditions preceding the flash flood that occurred the 18th May 2015 in Salgar (Colombia). It gathers information not only from observations but also from models or Synoptics patterns which could help to classify similar situations in future events. This information is clearly presented and allows the reader to follow the different factors that lead to that terrible event. I think the paper deserves publication due to the amount of information and valuable conclusions or lessons that can be extracted from it. I have some minor comments that could be addressed before its publication.

The figures might need some improvement to make them more understandable, most

of the comments about the figures have been already addressed by other reviewers. I would recommend to include the time of the event in temporal figures such as Fig 10e and Fig 10f. It could also help in Figure 14 but it is not that necessary.

The concept of flash-flood is usually defined as: "a flood of which peak appears within, in general, six hours from the onset of torrential rainfall" (NOAA). I would recommend introducing this definition at the beginning of the paper (Pag 2, l 35) and try to distinguish between the preconditioning raining events (This could be the main events before 17th May at night) and the rainfall that actually caused the flash-flood event.

Pag 6, l135-136: The "high" correlation between the QPE and observations does not guarantee high reliability of the derived radar precipitation. In fact, any bias would not affect the correlation. Besides, in your Figure 2 can be observed a clear overestimation of the radar QPE (Surprising taking into account sentence l325 about rainfall underestimation). Try to remove this sentence or make it clear what is your real point.

Pag 7 l152-154: Please, rephrase this ending. It is unclear.

Pag 7 l 158: It is mentioned the satellite for issuing warnings to the public. This comment is fine but I am surprised there is no comment about the capability of both Satellite (and mostly radar) for providing early warning for rainfall events by using extrapolation techniques or cell tracking techniques. Maybe it is worthy to include some information about these techniques. I understand that testing these techniques are out of the scope of this paper but still, a comment may help future readers.

Pag 12 l 287: "Flash-floods ... are more likely during the MAM quarter. Reference or statistic proving that sentence?

Pag 26 l 470-471. 5-years radar record is not enough to reach strong conclusions. This point has been addressed by the other reviewer. I recommend to introduce it here as a weak point fo your study.

WRF section. It is proved that WRF had a good potential in the given case but it is

known that the orography plays a significant role in precipitation in WRF model and in this case, so it could be that WRF is quite often producing a similar pattern over the mountaneos regions. Consequently, to give predictive value to WRF for flash flood, other events were similar synoptic patterns but not flash flood occurred should be introduced in the section and prove WRF did not predict similar patterns or intensities.

Discussion and conclusion section. I am missing some hydrological information of the basin, such as the peak time of the basin, and also some information about the required lead-time required for mitigation of the flash-flood in the studied region. I think this is an important matter when trying to investigate conditions to alert for future flash-flood in the region. And also, an explanation of how to derive this lead-time required. The combination of these two times (peak and warning) is really important, in some large basins, the peak time is so long that observations are enough to issue warning systems. In other, nowcasting can do the work while in other is so short that NWP models or synoptic factors are required for this warning to be effective. I think this is important to be introduced in the paper.

---

## Author Comment (AC2) · 18 Sep 2019

We sincerely thank you for the insightful suggestions to improve the manuscript as well as for your kind comments. We are in the process of addressing each of the points made by both reviewers. Regarding the figures, we made the modifications suggested by reviewer one and also modified new Figures 11 and 15 following your suggestions.

In new Figures 11 and 15: The gray shadow corresponds to the approximate timing of the flash flood event in different parts of the basin according to local authorities and the community (from 02:10 to 02:40 LT) - As reported by the media and the national government:

[Figure]

http://www.elcolombiano.com/antioquia/tragedia-en-antioquia-salgar-un-ano-despues-XX4145514

https://caracol.com.co/emisora/2015/12/25/medellin/$1451076926_792470.html$

http://portal.gestiondelriesgo.gov.co/Paginas/Noticias/2015/Antecion-Emergencia-Salgar-Antioquia.aspx
* * *
[Figure]

**Fig. 1.** New Figure 11

[Figure]

**Fig. 2.** New Figure 15

---

## Author Comment (AC3) · 28 Sep 2019

**Response to Reviewers**

September 27, 2019

**Manuscript title:**  Meteorological Conditions Leading to the 2015 Salgar Flash Flood: Lessons for Vulnerable Regions in Tropical Complex Terrain

**Authors:** Carlos D. Hoyos, Lina I. Ceballos, Jhayron S. Pérez-Carrasquilla, Julián Sepulveda, Silvana M. López, Manuel D. Zuluaga, Nicolás Velásquez, Laura Herrera-Mejía, Olver Hernández, Gisel Guzmán, Mauricio Zapata.

**Response to Reviewer #1**

The paper presents an overview of the meteorological conditions that triggered the flash flood in Salgar (Colombia) in May 2015. The authors put together a lot of information using radar imagery, satellite retrievals, atmospheric modelling and trajectories. . . The combination of different sources allowed a comprehensive overview of the case study.

The presentation of methods and sources is clear and relevant. Figures are useful and relevant. Captions are detailed and full of required information to understand the figure. The presentation of results is also relevant showing that the floods results in a complex scenario associating previous huge rainfalls in the upstream basin and triggering rainfalls 3 days after although for this third episode, hourly precipitations was not so extreme.

We thank the reviewer for the kind words and the insightful comments. We have considered all the suggestions and addressed all the questions as it is shown in the following sections.

**Response to comments**

1 Two figures would require some modifications in my opinion :

- Figure 1a is small and we are not able to locate the city of Salgar the main urban areas hit by the 18th may Flash floods in the frame of the catchment.
  We agree with the reviewer. We have modified the figure as follows:

- Figure 3 : rainfall amount usually are represented with bars instead of curves (curves are indeed reserved for cumulative precipitations
  We have also modified Figure 3 in the manuscript. See Figure 2:

- Fig4f 4g ,SST are not really visible.
  We agree with the reviewer. We decided to change Figure 4, making it two different figures: Panels f and g in the original version of Figure 4 were deleted (see Figure fig:climatology in this document), and we included in the manuscript a new figure showing

[Figure]

Figure 1: (a) to (c) Geographical context of the municipality of Salgar and La Liboriana basin. The panels present (a) the location of Colombia, the Department of Antioquia, and (b) and (c) Salgar, settled in the southwestern region of the Department, on the eastward facing hill of the western branch of the Andes Cordillera. The circular gray shadow denotes the 120 km radius area from the weather radar (C-Band) site. (b) Three-dimensional representation of La Liboriana basin using ALOS-PALSAR Digital Elevation Model (12.5 m resolution) highlighting the steepness of the region. Cerro Plateado corresponds to the highest hills of the basin at 3600 m.a.s.l. (c) Zoom to the municipality of Salgar and La Liboriana basin, showing the location of four rain gauges (1 to 4) from the Colombian National Weather Service (IDEAM), and two rain gauges (a and b) installed post-event by the Sistema de Alerta Temprana de Medellín y el Valle de Aburrá (SIATA). The IDEAM gauges correspond to codes 11020010 (1), 11025010 (2), 26210110 (3), and 26215010 (4). (d) y (e) Images before and after the Salgar May 18, 2015 flash flood event, showing an example of landslide scars in the upper basin, and erosion and changes in the delineation of the main channel. The first column corresponds to the pansharpened natural color composition from Landsat 8 images (path: 009, row: 056) with 15 m resolution for the upper basin ( Before: 2013-07-16, and After: 2017-05-24). The second column corresponds to Google Earth Pro V 7.3.2.5491, 2018 for the main channel. The images are from Digital Globe, 2018 (Before: 2014-10, After: 2015-05).

[Figure]

Figure 2: Daily precipitation during May 2015 near Salgar, recorded at the IDEAM in situ rain gauges shown in Figure 1. Bars correspond to daily cumulative precipitation and the dashed lines to the evolution of the cumulative precipitation during May 2015.

[Figure]

Figure 3: a) Mean annual cycle of precipitation near Salgar from IDEAM rain gauges and TRMM 3B42 product (thick yellow line) using data from 1998 to 2018. The panel also shows the 5-year average monthly QPE rainfall from 2014 to 2018. b) Scatter plot between monthly TRMM records and one of the in situ gauges, showing the Pearson correlation between both series (0.64). c) Precipitation anomalies over La Liboriana, after filtering all variability with periods equal to or shorter than 13 months. d) 10-year windowed moving correlation between the Multivariate ENSO Index (MEI) and in situ precipitation. e) Spatial distribution of the correlation between filtered SST and in situ precipitation.

the cumulative precipitation and mean SST during May 2015, and their anomalies, all in separate panels to avoid the overlapping (see Figure fig:climatology2 in this document).

2 I suggest to replace hydrometeorological in the title by meteorological as there is no information in the paper on the hydrological characteristics. If authors want to keep the title "hydrometeorological " in the title they should introduce information on hydrology e.g. in the discussion comparing the occurrence of peak discharge and the highest hourly precipitations

Before submitting the manuscript, we had the same discussion among the authors of the manuscript. We agree the document is mostly about the meteorological conditions, but we had decided initially to use the term "hydrometeorological" in the first version of the manuscript because of the analysis presented in section 3.3 on the first-order hydrometeorological processes associated with the event. In the mentioned section, we assess the spatial distribution of the extreme hourly cumulative precipitation percentiles for La Liboriana basin, relative to

[Figure]

Figure 4: f) Regional cumulative precipitation from the TRMM 3B42 product (blue-to-red shading) and mean SST (green-to-white shading and black lines) during May 2015. g) Cumulative precipitation (red-to-white shading) and mean SST (green-to-pruple shading and black lines) anomalies relative to the long-term May conditions. Temperature contours are every 0.5°C

the basin slope and location; this analysis falls in the area of hydrometeorology. Considering the comments by the reviewer, and the fact that the paper is already long in its current form, we have decided to change the title to "meteorological conditions...". Also, most of the hydrological considerations are included in Velásquez et al. (2018). Nevertheless, we included in the discussion, a comparison between the occurrence of the peak discharge and the timing of the maximum precipitation intensity. See the following comments:

Regarding the lead times, in the case of La Liboriana, an analysis of the lag between the peak discharge time relative to the maximum intensity for the events presented in Velásquez et al. (2018) suggests the minimum useful forecast lead time is approximately 1.5 hours. This lead time also matches the estimates of the time of concentration of La Liboriana basin (between 1.3-1.7 hours) using different methodologies, including the **?** and Giandotti (as cited in (**?**)) equations.

3 Discussion and conclusion The discussion is rather a conclusion. Some comparison with other cases would be welcome. On another hand, it is relevant to consider not only the rainfall of the 18th of May but also the cumulative rainfalls of the previous days but (p.28) the 96-hours cumulative precipitation forecast is suggested to be the best period to assess the potential occurrence of flash floods but this conclusion is valid only for this case study. In another case, the relevant period could be 100 hours or 108 hours etc. flood risk managers in Salgar may have at their disposal a running cumulative amount of rainfall over the basin for any period and above all, an updated assessment of soil moisture over the basin. But that is matter of hydrologists.

We followed the suggestion by the reviewer and we have added a comparison with other cased in the literature, the comment about the useful lead-time, and a sentence about the 96-hour

cumulative precipitation.

4 One of the weaknesses of the paper is the shortness of the reference period to estimate the return period of the 2015 rainfall event. To conclude that the flash flood is the more intense over the last five years is not really interesting. Authors account for the radar QPE record over the period 2014-2018. We suppose that they have not any other long series at their disposal. The discussion could introduce some qualitative testimonies about previous flash floods (last century?) that would allow putting the 2015 event in perspective.

We agree with both reviewers. We have rephrased the following parts of the corresponding section to highlight the limitations and the lack of robustness of the conclusions regarding the recurrence rate of the extreme event, as follows:

One of the goals of this work is to assess the likelihood of occurrence of extreme events similar to the one, or to the ones, triggering La Liboriana flash flood. In other words, it is important to evaluate whether or not the characteristics of the May 18, 2015 flood were exceptional, and ideally, their recurrence rate. In a traditional sense, it would be desirable to estimate a return period of the conditions that led to the la Liboriana flash flood. However, the length of the historical radar QPE record is not enough for a robust estimation of the return period. Considering this limitation, the previous analysis together with first-order hydrometeorological considerations allow us to conduct a preliminary assessment of the exceptionality of the precipitation conditions associated with the event.

The probabilities presented in this paragraph are not robust as they have been estimated based on a 5-year radar record; not enough for assessing extreme event recurrence as mentioned previously, however, the practical implications are important as the results suggest that La Liboriana event was, in fact, exceptional compared to all events in the 5-year radar record, but in particular, in the upper part of the basin, implying that for optimal risk management it is necessary to consider the spatial distribution of cumulative rainfall relative to the geomorphological features of the basin. In other words, while the individual event on May 18 was not exceptional, the climatological anomalies were negative-to-normal, and the synoptic patterns around the extreme event were similar to the expected ones for the region, the combination of high rainfall accumulation as a result of successive precipitation events over the basin, followed by a moderate extreme event is unique in the available observational record. The evidence for the La Liboriana basin also suggests that the 96-hour period is more appropriate to analyze the extreme event (Figures ??c and d). In this case, for the upper part of the basin, there is no other event in the historical record with one-hour and 96-hour cumulative values larger or equal to the Salgar event. An analysis of the historical disaster records available at https://www.desinventar.org for the Salgar, and the analysis and historical accounts presented in ? and ?, there have been 11 extreme flash floods and torrential floods from from 1922 to 2019, five of them with fatalities. According to these reports, the largest so far corresponds to the event during Mary 18, 2015. The deadliest most event before the flash flood assessed in this study corresponds to a flash flood and torrential flow during June 1971, killing 45 persons.

5 typing mistakes:

  – Line 241 : would not the temperature be $27^o$-$29^o$ instead of $37^o$-$39^o$?
    The reviewer is right. The correct temperature range is $27^o$-$29^o$.

  – Line 271 : It is important to note that the we use moisture may reach the atmospheric column over La Liboriana at different levels.
    We have corrected the sentence as follows: It is important to note that moisture may reach the atmospheric column over La Liboriana at different levels.

- l. 272 : reaching

  The typo has been fixed.
- l. 493 : specific

  The typo has been fixed.
- p. 30 caption of figure 19 : "topography" and "The color table for relative humidity corresponds to panels.."

  The typo was fixed and the sentence has been rewritten.
- l. 517 : suggests

  The typo has been fixed.
- L. 528 : additionally

  The typo has been fixed.

**Response to Reviewer #2**

The current manuscript deeply studies the different meteorological conditions preced- ing the flash flood that occurred the 18th May 2015 in Salgar (Colombia). It gathers in- formation not only from observations but also from models or Synoptics patterns which could help to classify similar situations in future events. This information is clearly presented and allows the reader to follow the different factors that lead to that terrible event. I think the paper deserves publication due to the amount of information and valu- able conclusions or lessons that can be extracted from it. I have some minor comments that could be addressed before its publication.

We thank the reviewer for the constructive and helpful comments during the review process. We think the manuscript is more solid and it will have a better impact in the community.

**0.1   Response to the comments:**

1 The figures might need some improvement to make them more understandable, most of the comments about the figures have been already addressed by other reviewers. I would recommend to include the time of the event in temporal figures such as Fig 10e and Fig 10f. It could also help in Figure 14 but it is not that necessary.

We have taken into consideration the suggestions by both reviewers and have modified the figures accordingly. See the modifications of former Figures 10 and 14 in the orogicanl manuscript in Figures 5 and 6 in this document.

2 The concept of flash-flood is usually defined as: "a flood of which peak appears within, in general, six hours from the onset of torrential rainfall" (NOAA). I would recommend introducing this definition at the beginning of the paper (Pag 2, l 35) and try to distinguish between the preconditioning raining events (This could be the main events before 17th May at night) and the rainfall that actually caused the flash-flood event.

We introduced the definition as suggested by the reviewer, as follows:

Flash floods are associated with short-lived, very intense convective precipitation events, usually enhanced by the orography, over highly saturated land surfaces with steep terrains (Šálek et al., 2006; Llasat et al., 2016; Douinot et al., 2016; Velásquez et al., 2018). According to the

[Figure]

Figure 5: 5-minute time series of convective and stratiform rainfall over the La Liboriana basin during the period of maximum accumulation of each of the three events in Figure **??**, as well as the evolution of the area of the basin covered by either stratiform or convective precipitation. a) and b) correspond to the May 14-15 event, c) and d) to the May 17 event, and e) and f) to the May 17-18 triggering event.

[Figure]

Figure 6: a) to c) Hovmöller diagrams (longitude-time cross-section) at 6°N for different radar scanning tilts at 1, 2 and 4°, respectively. d) Topography of the region at 6°N. e) Continuous lines show the temporal evolution of $Zh$ along the black line in a) to c), and the dashed line shows the corresponding terrain elevation below the precipitating cloud.

US National Oceanic and Atmospheric Administration (NOAA), flash floods are triggered by heavy or excessive rainfall in a short period of time, appearing, in general, within six hours from the onset of torrential rainfall (Jha et al., 2012). However, it is important to state that flash floods are not only the result of the rainfall event immediately before the flooding (the triggering event); the spatio-temporal structure of the rainfall in the days prior to the extreme flooding (preconditioning events) also play an important role modulating the overall moisture in the basin and in the occurrence of flooding. Flash floods are highly destructive, often resulting in significant human and economic losses, making them one of the most catastrophic natural hazards (Jonkman, 2005; Roux et al., 2011; Gruntfest and Handmer, 2001). Jonkman (2005), based on information from the International Disaster Database, shows that between 1975 and 2001 a total of 1816 worldwide freshwater flood events killed over 175 thousand people and affected more than 2.2 billion people. These events not only caused human and economic losses but also damages to ecosystems and loss of historical and cultural values. In Colombia, there have been several flash flood events in the last decade associated with large-scale climate forcing patterns and with isolated extreme precipitation events. The 2010-2011 La Niña event triggered more than 1200 flooding events, affecting the lives of more than 3 million people (around 7% of the country's population) and causing damages estimated in more than 6.5 billion US dollars (UN-CEPAL, 2012).

3 Pag 6, l135-136: The "high" correlation between the QPE and observations does not guarantee high reliability of the derived radar precipitation. In fact, any bias would not affect the correlation. Besides, in your Figure 2 can be observed a clear overestimation of the radar QPE (Surprising taking into account sentence l325 about rainfall underes- timation). Try to remove this sentence or make it clear what is your real point.

The sentence in l325 in the original version of the manuscript is a general statement about radars and QPE. As noted by the reviewer, that is not necessarily true in our case. For this reason, we deleted the sentence. Also, we added in the document statistics not only about the correlation observed in Figure 2, but also about the root mean squared error and the mean absolute error, as follows:

Figure 2 shows a correspondence between the hourly and daily cumulative precipitation estimated using the QPE technique and the precipitation registered in situ. The correlation between the hourly and daily rain gauge records and the QPE estimations are, respectively, 0.65 and 0.74. Similarly, the root mean square error and the mean absolute error for the hourly and the daily records are 3.8 and 2.2 mm, and 10.4 and 7.2 mm, respectively. The high correlations, and relatively small hourly errors indicate, despite the evident overestimation relative to the in situ tipping-bucket gauge records (slopes in Figure ?? are 0.6 and 0.62), a high reliability of the derived precipitation based on radar reflectivity fields.

4 Pag 7 l152-154: Please, rephrase this ending. It is unclear.

The sentenced was rephrased following the reviewer's suggestion: We estimated the velocity fields from precipitation retrievals (QPE technique) using a pattern cross-correlation technique and a variational approach, to satisfy continuity, for all the precipitation events over La Liboriana from May 2014 to August 2018. The dates of the precipitation events were obtained from the radar record. Precipitation events with an average rainfall intensity less than than 0.2 mm h$^{-1}$ were not considered.

5 Pag 7 l 158: It is mentioned the satellite for issuing warnings to the public. This comment is fine but I am surprised there is no comment about the capability of both Satellite (and mostly radar) for providing early warning for rainfall events by using extrapolation techniques or cell

tracking techniques. Maybe it is worthy to include some information about these techniques. I understand that testing these techniques are out of the scope of this paper but still, a comment may help future readers.

We agree with the reviewer; we have introduced the following comment in the manuscript: Although it is out of the scope of this research, it is important to highlight that extreme convection early detection and tracking algorithms based on combined satellite and radar information might also play a decisive role in risk management, anticipating the threat to regions of interest. The reader is referred to the description of skillful Lagrangian tracking algorithms such as the ones described in Carvalho and Jones (2001); Handwerker (2002); Vila et al. (2008); Zinner et al. (2008); Bellerby et al. (2009); Kober and Tafferner (2009); Merk and Zinner (2013).

6 Pag 12 l 287: "Flash-floods ... are more likely during the MAM quarter. Reference or statistic proving that sentence?

The sentence in the original manuscript is a suggestion derived from the results: **...This feature is relevant as it suggests that flash floods and rainfall-triggered landslides are more likely during the MAM quarter.** However, the reviwer is right regarding the need for a convincing statistic result. We have added the following comment in the manuscript:

The previous assessment is validated by the analysis of the historical disaster records available at https://www.desinventar.org, for the sub-region of the Department of Antioquia including the municipality of Salgar (the southwest region). The number of flash floods and torrential flows in the sub-region, from 1922 to 2019, are 6 during DEF, 25 during MAM, 13 during JJA, and 10 during SON.

7 Pag 26 l 470-471. 5-years radar record is not enough to reach strong conclusions. This point has been addressed by the other reviewer. I recommend to introduce it here as a weak point fo your study.

We agree with both reviewers. We have rephrased the following parts of the corresponding section to highlight the limitations and the lack of robustness of the conclusions regarding the recurrence rate of the extreme event, as follows:

One of the goals of this work is to assess the likelihood of occurrence of extreme events similar to the one, or to the ones, triggering La Liboriana flash flood. In other words, it is important to evaluate whether or not the characteristics of the May 18, 2015 flood were exceptional, and ideally, their recurrence rate. In a traditional sense, it would be desirable to estimate a return period of the conditions that led to the la Liboriana flash flood. However, the length of the historical radar QPE record is not enough for a robust estimation of the return period. Considering this limitation, the previous analysis together with first-order hydrometeorological considerations allow us to conduct a preliminary assessment of the exceptionality of the precipitation conditions associated with the event.

The probabilities presented in this paragraph are not robust as they have been estimated based on a 5-year radar record; not enough for assessing extreme event recurrence as mentioned previously, however, the practical implications are important as the results suggest that La Liboriana event was, in fact, exceptional compared to all events in the 5-year radar record, but in particular, in the upper part of the basin, implying that for optimal risk management it is necessary to consider the spatial distribution of cumulative rainfall relative to the geomorphological features of the basin. In other words, while the individual event on May 18 was not exceptional, the climatological anomalies were negative-to-normal, and the synoptic patterns

around the extreme event were similar to the expected ones for the region, the combination of high rainfall accumulation as a result of successive precipitation events over the basin, followed by a moderate extreme event is unique in the available observational record. The evidence for the La Liboriana basin also suggests that the 96-hour period is more appropriate to analyze the extreme event (Figures **??**c and d). In this case, for the upper part of the basin, there is no other event in the historical record with one-hour and 96-hour cumulative values larger or equal to the Salgar event. An analysis of the historical disaster records available at https://www.desinventar.org for the Salgar, and the analysis and historical accounts presented in **?** and **?**, there have been 11 extreme flash floods and torrential floods from from 1922 to 2019, five of them with fatalities. According to these reports, the largest so far corresponds to the event during Mary 18, 2015. The deadliest most event before the flash flood assessed in this study corresponds to a flash flood and torrential flow during June 1971, killing 45 persons.

8 WRF section. It is proved that WRF had a good potential in the given case but it isknown that the orography plays a significant role in precipitation in WRF model and in this case, so it could be that WRF is quite often producing a similar pattern over the mountaneos regions. Consequently, to give predictive value to WRF for flash flood, other events were similar synoptic patterns but not flash flood occurred should be in- troduced in the section and prove WRF did not predict similar patterns or intensities.

The point raised by the reviewer is important. We added a Figure as an appendix to show that this is not the case, and that there is important information in the WRF forecas useful for risk management. We briefly discussed this issue in the manuscript as follows:

In order to evaluate the actual usefulness of WRF in flash flood likelihood assessment, it is important to discard the possibility that the WRF-simulated rainfall pattern associated with La Liboriana flash flood (Figure 20c) is an artifact of the model, overemphasizing the orographic enhancement of precipitation. During the April-May 2015 rainy season, radar observations show a total of 57 rainfall events in addition to the ones leading to La Liboriana. Among all 57 events, 42 show a similar synoptic pattern. On the other hand, WRF simulations of the 42 events, from the daily operational SIATA weather forecasts, do not exhibit extreme orographic intensification (see Figure 7 for the spatial pattern of the WRF-simulated cumulative precipitation for the top nine rainfall events our of the mentioned 42 cases). This suggests that the WRF simulation during the period of La Liboriana is not an artifact of the model, providing useful information for risk management.

9 Discussion and conclusion section. I am missing some hydrological information of the basin, such as the peak time of the basin, and also some information about the required lead-time required for mitigation of the flash-flood in the studied region. I think this is an important matter when trying to investigate conditions to alert for future flash- flood in the region. And also, an explanation of how to derive this lead-time required. The combination of these two times (peak and warning) is really important, in some large basins, the peak time is so long that observations are enough to issue warning systems. In other, nowcasting can do the work while in other is so short that NWP models or synoptic factors are required for this warning to be effective. I think this is important to be introduced in the paper.

We agree with the reviewer. We have added the following comment:

Regarding the lead times, in the case of La Liboriana, an analysis of the lag between the peak discharge time relative to the maximum intensity for the events presented in Velásquez et al. (2018) suggests the minimum useful forecast lead time is approximately 1.5 hours. This lead time also matches the estimates of the time of concentration of La Liboriana basin (between

1.3-1.7 hours) using different methodologies, including the **?** and Giandotti (as cited in (**?**)) equations.

**References**

[revised manuscript text omitted]

---

## Author Comment (AC4) · 28 Sep 2019

See attached document including the response to both reviewers.

Please also note the supplement to this comment:
https://www.nat-hazards-earth-syst-sci-discuss.net/nhess-2019-171/nhess-2019-171-AC4-supplement.pdf
* * *

---

## Author Comment (AC6) · 28 Sep 2019

See modified manuscript as a supplement

Please also note the supplement to this comment:
https://www.nat-hazards-earth-syst-sci-discuss.net/nhess-2019-171/nhess-2019-171-AC6-supplement.pdf
* * *